# SECOND-ORDER FORWARD-MODE AUTOMATIC DIFFERENTIATION FOR OPTIMIZATION

## ABSTRACT

Forward gradient methods offer a promising alternative to backpropagation. Optimization that only requires forward passes could simplify hardware implementation, improve parallelism, lower memory cost, and allow for more biologically plausible learning models. This has motivated recent forward-mode automated differentiation (AD) methods. This paper presents a novel second-order forward-mode AD method for optimization that generalizes a second-order line search to a $K$-dimensional hyperplane. Unlike recent work that relies on directional derivatives (or Jacobian–Vector Products, JVPs), we use hyper-dual numbers to jointly evaluate both directional derivatives and their second-order quadratic terms. As a result, we introduce forward-mode weight perturbation with Hessian information for K-dimensional hyper-plane search (FoMoH-$K$D). We derive the convergence properties of FoMoH-$K$D and show how it generalizes to Newton's method for $K = D$. We demonstrate this generalization empirically, and compare the performance of FoMoH-$K$D to forward gradient descent (FGD) on three case studies: Rosenbrock function used widely for evaluating optimization methods, multinomial logistic regression with 7,850 parameters, and learning a CNN classifier with 431,080 parameters. Our experiments show that FoMoH-$K$D not only achieves better performance and accuracy, but also converges faster, thus, empirically verifying our theoretical results.

## 1 INTRODUCTION

Backpropagation, that is, reverse-mode automated differentiation (AD), is widely used for optimization, particularly in machine learning, despite the additional computational complexity and memory requirement associated with the need for a backward pass. As a result, there is a growing interest in investigating the practical plausibility of forward-mode AD as a more efficient way to estimate gradients with the added interest of being relevant to biological learning mechanisms. There has been significant progress towards this goal and the recently proposed approach for forward gradient descent (FGD) (Baydin et al., 2022) relies on sampling tangent vectors to update function parameters. These parameters are updated by subtracting the tangents that are scaled by their directional derivatives. As a result, their approach only requires forward passes, and avoids the computation and memory costs associated with implementing the backward pass of reverse-mode AD. The current challenges of working with FGD have been on reducing the variance of the gradient estimator with an increase of dimensions (Ren et al., 2022; Fournier et al., 2023) and improving on the performance and convergence rate of optimization.

In this paper, we address these challenges by using Hessian information to develop a second-order forward-mode AD approach for optimization without any backpropagation (BP). While second-order approaches have been used in BP, our paper is the first to develop a second-order forward-mode AD approach. Second-order derivative information provides optimization routines with information about the local curvature. However, computing the full Hessian for high-dimensional problems requires prohibitively large memory. Instead of building the full Hessian, we introduce a new second-order approach that leverages a partial Hessian. Our approach, FoMoH-$K$D, builds a $K \times K$ Hessian in a sub-space of the full function space. As a result, FoMoH-$K$D is a forward-mode AD approach that outperforms the previous forward-mode-only first-order gradient descent method, FGD. We demonstrate and prove that our approach bridges the gap between a line search and a full Newton step. By incorporating Hessian information, FoMoH-$K$D leverages second-order

derivatives to provide curvature approximations of the objective function. This allows the method to adjust the step size and direction with greater precision compared to a simple line search, which only uses gradient information. Simultaneously, FoMoH-$K$D avoids the computational complexity of a full Newton step, which requires the matrix vector multiplication of the inverse Hessian and the gradient. Our approach balances the efficiency of a line search with the optimization effectiveness and accuracy of a full Newton method, offering a robust and versatile optimization technique.

In summary, the central contributions of this paper are as follows:

- **Approach:** We introduce FoMoH-$K$D, a novel forward-mode-only $K$-dimensional hyperplane search that uses second-order information to build partial Hessians. Our approach enables a natural trade-off between computational complexity and optimization performance.

- **Theory:** We show and prove that FoMoH-$K$D generalizes to Newton's method, without the need for backpropagation. We also compare the convergence rate of FGD to FoMoH-$K$D and show that, under certain assumptions (§5), the expected reduction in error from the minimum decays at an exponential rate of $(D - K)/D$. This compares with the previous approach of FGD, which has an expected sub-linear convergence rate.

- **Experiments:** We demonstrate FoMoH-$K$D on optimization problems of different parameter sizes and difficulty to show the advantage of second-order information over the first-order forward-mode-only approach of FGD.

- **Open-Source Code:** We release an AD backend in PyTorch that implements nested forward AD and interfaces with PyTorch models.

The rest of the paper is organized accordingly: §2 and §3 summarize the related work and relevant preliminary information regarding AD, which is then followed by §4 that summarizes the extension of forward-mode AD to higher order derivatives. §5 introduces FoMoH-$K$D and the convergence analysis. Finally, §6 provides experimental results that explore the behavior of FoMoH-$K$D. We then conclude in §7. The appendix contains additional results and formal proofs.

## 2 RELATED WORK

There has been considerable interest in developing approaches that avoid reverse-mode AD and its backward pass. In moving away from BP, it might be possible to build optimization algorithms that more closely align with biological systems (Bengio et al., 2015; Hinton, 2022), or enable neural networks to run on emerging hardware architectures, such as analog optical systems (Pierangeli et al., 2019). Baydin et al. (2022) introduced FGD as a possible replacement for BP in neural network training by relying on weight perturbations. This removed the truncation error of previous weight perturbation approaches (Pearlmutter, 1994) by moving to forward-mode AD. While FGD is a promising BP-free approach, the scaling of FGD to high-dimensional models is challenging due the variance of the gradient estimator (see derivation in §B.3). This challenge has led to multiple efforts that have focused on reducing this variance (Ren et al., 2022; Silver et al., 2021; Fournier et al., 2023). In particular, a common approach has been to rely on local BP steps to reduce the variance and/or to provide a better guess for the perturbation direction. In our work, we focus on second-order forward-mode optimization approaches, but highlight that these other approaches on variance reduction are orthogonal to our approach and could also be combined together and generalized to FoMoH-$K$D.

Becker et al. (1988) were one of the first to the use second-order information to optimize neural networks. This approach leverages a local BP step (Le Cun, 1987) to capture the diagonal Hessian terms resulting in what they call a "pseudo-Newton step". Both LeCun et al. (1998b, §6–§9) and Goodfellow et al. (2016, §8.6) provide discussions of the use of Hessian information for neural network training and cover Newton's method, as well as the Levenberg–Marquardt algorithm (Levenberg, 1944; Marquardt, 1963), conjugate gradients, and BFGS. A key objective of these approaches is to investigate whether second-order information can be leveraged for neural network training without the prohibitively high cost of a full Hessian evaluation. Additionally, the research question then explored is whether the approximated Hessian is still good enough to give the desired advantage over first-order methods. Examples of effective approaches often rely on diagonal preconditioners, with some approximating or directly using the diagonal of the Hessian (Yao et al., 2021; Le Cun, 1987; Becker et al., 1988), and others leveraging momentum and a variant of the diagonal

of the empirical Fisher (Duchi et al., 2011; Tieleman & Hinton, 2012; Kingma & Ba, 2015). While extending a gradient preconditioner to the full inverse Hessian is generally infeasible, there are approaches that use: block-diagonal approximations, low-rank approximations, and Krylov-subspace based approximations (Le Roux & Fitzgibbon, 2010; Vinyals & Povey, 2012). Additional references for these preconditioners can be found in Martens (2020). Finally, although it is challenging to scale second-order approaches to large neural network models, a recent work, Sophia (Liu et al., 2023), has managed to show success in using such an approach. Like previous works, they use a diagonal approximation of the Hessian. Importantly, they show that Sophia requires half the number of update steps as Adam (Kingma & Ba, 2015) to train a language model (GPT-2 (Radford et al., 2019)). It is worth noting that none of the described second-order optimization methods rely solely on forward-mode AD like the one being proposed in this paper.

## 3 AUTOMATIC DIFFERENTIATION

In this section we summarize the common definitions of forward-mode and reverse-mode automatic differentiation. For a complete introduction, please refer to Griewank & Walther (2008). We use $\mathbf{x} \in \mathbb{R}^D$ to denote a column vector.

**Forward-Mode AD** (Wengert, 1964) applies the chain rule in the *forward* direction. The forward-mode evaluation, $F(\boldsymbol{\theta}, \mathbf{v})$, requires an additional tangent vector, $\mathbf{v} \in \mathbb{R}^D$, along with the parameter vector $\boldsymbol{\theta} \in \mathbb{R}^D$ for a function $\boldsymbol{f} : \mathbb{R}^D \to \mathbb{R}^O$. The result of the evaluation, $[\boldsymbol{f}(\boldsymbol{\theta}), \nabla \boldsymbol{f}(\boldsymbol{\theta})\mathbf{v}]$, is a function evaluation and the corresponding Jacobian vector product (JVP), where $\nabla \boldsymbol{f}(\boldsymbol{\theta}) \in \mathbb{R}^{O \times D}$. For a unidimensional output function, the JVP is the directional derivative, $\nabla \boldsymbol{f}(\boldsymbol{\theta}) \cdot \mathbf{v}$. The time and space (memory cost) complexity are linear, both approximately twice that of a single function evaluation.[1] A common implementation of forward-mode AD is to use dual numbers. A dual number $a + b\epsilon \in \mathbb{D}(\mathbb{R})$ contains a real (primal) component, $a \in \mathbb{R}$, and a dual component, $b \in \mathbb{R}$. We can think of this as representing a truncated Taylor series, $a + b\epsilon + \mathcal{O}(\epsilon^2)$, notationally simplified by the rule $\epsilon^2 = 0$. Using this, $f(a + b\epsilon) = f(a) + \nabla f(a)b\epsilon$. A simple example can be shown for the function, $f(a_1, a_2) = a_1 \times a_2$. Using dual numbers, $(a_1 + b_1\epsilon)(a_2 + b_2\epsilon)$, we retrieve the function evaluation and the product rule: $a_1 a_2 + (a_1 b_2 + b_1 a_2)\epsilon$. This can be extended to multiple dimensions, and is the basis of forward-mode AD: lift all real numbers $\mathbb{R}$ to dual numbers $\mathbb{D}(\mathbb{R})$.

**Reverse-Mode AD** requires both a forward pass and a reverse pass. The reverse-mode evaluation $R(\boldsymbol{\theta}, \boldsymbol{u})$, also requires the additional adjoint vector, $\boldsymbol{u} \in \mathbb{R}^O$, which is often set to 1 for scalar-valued functions. Using the same notation as for forward-mode, an evaluation of reverse mode results in the vector-Jacobian product, $\boldsymbol{u}^\top \nabla \boldsymbol{f}(\boldsymbol{\theta})$, as well as the function evaluation. When $\boldsymbol{u} = 1$, this results in the gradient $\nabla \boldsymbol{f}(\boldsymbol{\theta})$. Reverse-mode is required to store intermediate values on the forward pass that are used during the reverse pass. This results in a higher time and space complexity, that is higher computational cost and memory footprint per call of $R(\boldsymbol{\theta}, \boldsymbol{u})$. However, for the scalar-valued functions ($O = 1$) that are common for ML optimization problems, only a single call of $R(\boldsymbol{\theta}, \boldsymbol{u})$ is needed to collect all the gradients compared to $D$ (dimension of inputs) calls of $F(\boldsymbol{\theta}, \mathbf{v})$. This is one of the key reasons for the widespread adoption of the reverse-mode AD in current gradient-based machine learning methods, despite the higher memory footprint.

## 4 HIGHER-ORDER FORWARD MODE AUTOMATIC DIFFERENTIATION

As described in the previous section, forward-mode automatic differentiation implementations can use dual numbers, $\boldsymbol{\theta} + \mathbf{v}\epsilon$. Dual numbers can be extended to truncate at a higher order, or to not truncate at all; and to allow nesting by supporting multiple distinct formal $\epsilon$ variables (Pearlmutter & Siskind, 2007). Specifically focusing on second-order terms is often referred to as hyper-dual numbers (for example in the Aeronautics and Astronautics community (Fike & Alonso)). A hyper-dual number is made up from four components, which is written as $\boldsymbol{\theta} + \mathbf{v}_1\epsilon_1 + \mathbf{v}_2\epsilon_2 + \mathbf{v}_{12}\epsilon_1\epsilon_2$. In the same manner that we look at imaginary and real parts of complex numbers, we can look at the first derivative parts of a hyper-dual number by inspecting the $\epsilon_1$ and $\epsilon_2$ components, and we can look at the second derivative by inspecting the $\epsilon_1\epsilon_2$ component. To understand how this formulation arises, we can introduce the definitions $\epsilon_1^2 = \epsilon_2^2 = (\epsilon_1\epsilon_2)^2 = 0$ and replace the Taylor series expansion of

---

[1]More precisely, the basic time complexity of a forward evaluation is constant $\in [2, 2.5]$ times that of the function call (Griewank & Walther, 2008).

a function, $f : \mathbb{R}^D \to \mathbb{R}$ around $\boldsymbol{\theta}$ with perturbation, $\mathbf{d} \in \mathbb{R}^D$, with an evaluation of a hyper-dual number:[2]

$$f(\boldsymbol{\theta} + \mathbf{d}) = f(\boldsymbol{\theta}) + \nabla f(\boldsymbol{\theta})\mathbf{d} + \frac{1}{2}\mathbf{d}^\top \nabla^2 f(\boldsymbol{\theta})\mathbf{d} + \cdots$$

$$f(\boldsymbol{\theta} + \mathbf{v}_1\epsilon_1 + \mathbf{v}_2\epsilon_2 + \mathbf{v}_{12}\epsilon_1\epsilon_2) = f(\boldsymbol{\theta}) + \nabla f(\boldsymbol{\theta})\mathbf{v}_1\epsilon_1 + \nabla f(\boldsymbol{\theta})\mathbf{v}_2\epsilon_2$$
$$+ \nabla f(\boldsymbol{\theta})\mathbf{v}_{12}\epsilon_1\epsilon_2 + \mathbf{v}_1^\top \nabla^2 f(\boldsymbol{\theta})\mathbf{v}_2\epsilon_1\epsilon_2 + \cdots.$$

An alternative but isomorphic view is to regard $a + b\epsilon_1 + c\epsilon_2 + d\epsilon_1\epsilon_2 = (a + b\epsilon_1) + (c + d\epsilon_1)\epsilon_2$ as an element of $\mathbb{D}(\mathbb{D}(\mathbb{R}))$, with subscripts to distinguish the inner vs outer $\mathbb{D}$s; from an implementation perspective, hyperduals can be regarded as inlining the nested structures into a single flat structure.[3]

## 4.1 IMPLICATIONS OF HYPER-DUAL NUMBERS FOR MACHINE LEARNING

A function evaluation with hyper-dual numbers takes an input vector, $\boldsymbol{\theta} + \mathbf{v}_1\epsilon_1 + \mathbf{v}_2\epsilon_2 + \mathbf{0}\epsilon_1\epsilon_2$, with the $\epsilon_1\epsilon_2$ part set to zero. A typical setting to get exact gradient and Hessian elements is to set $\mathbf{v}_1 = \mathbf{e}_i$ and $\mathbf{v}_2 = \mathbf{e}_j$, where $\mathbf{e}_i$ and $\mathbf{e}_j$ are each a basis of one-hot unit vectors. Therefore, these basis vectors select the corresponding elements of the gradient and Hessian:

$$f(\boldsymbol{\theta}) + \nabla f(\boldsymbol{\theta})_i\epsilon_1 + \nabla f(\boldsymbol{\theta})_j\epsilon_2 + \nabla^2 f(\boldsymbol{\theta})_{ij}\epsilon_1\epsilon_2 = f(\boldsymbol{\theta} + \mathbf{e}_i\epsilon_1 + \mathbf{e}_j\epsilon_2 + \mathbf{0}\epsilon_1\epsilon_2).$$

A single loop over the input dimension provides the exact gradient, whereas a nested loop provides the full Hessian. As a side note, a single loop also can give the Hessian vector product. This is done by setting one of the tangent vectors to the chosen vector, and looping through the basis for the other tangent vector.[4] However, the key advantage is that with a single forward pass, we get both first order and second order information in the form of directional derivatives and the local curvature information respectively. Since we have already seen from Baydin et al. (2022) and follow up works (Ren et al., 2022; Fournier et al., 2023) that first-order forward-mode-only optimization routines can successfully train machine learning models, we now look to ask the question as to whether forward passes that provide second-order information can be used to further improve on current forward-mode AD optimization. Therefore, in this paper we investigate whether the additional access to local curvature through forward-mode AD can enable improvements over the current FGD algorithm.

**Local Curvature:** $\mathbf{v}_1^\top \nabla^2 f(\boldsymbol{\theta})\mathbf{v}_2$**.** The Hessian contains the curvature information at a point, $\boldsymbol{\theta}$, in the form of the second order partial derivatives. When we evaluate a function over hyper-dual numbers we get the bilinear form, $\mathbf{v}_1^\top \nabla^2 f(\boldsymbol{\theta})\mathbf{v}_2$. This is a function that operates over two vectors, such that it is linear in each vector separately. The bilinear form is common in optimization routines, such as for conjugate gradient descent to assess whether two vectors are conjugate with respect to the Hessian. The value of $\mathbf{v}_1^\top \nabla^2 f(\boldsymbol{\theta})\mathbf{v}_2$ tells us about how curvature covaries along the two vectors. In the case where $\mathbf{v}_1 = \mathbf{v}_2 = \mathbf{v}$, we arrive at the quadratic form that describes the curvature, which indicates the rate of change of the slope as you move in the direction of $\mathbf{v}$. The quadratic form also provides information on the convexity of the function at that point. If $\mathbf{v}^\top \nabla^2 f(\boldsymbol{\theta})\mathbf{v} > 0, \forall \mathbf{v} \in \mathbb{R}^D_{>0}$, then the function is convex at that point and the Hessian is positive definite. The curvature also indicates the sensitivity of moving in certain directions. For example, when using gradients to optimize a function, taking a gradient step in a region of low curvature (with a small learning rate) is likely to increase the value of the function. However the same sized step in a region of large curvature could significantly change the value of the function (for better or worse).

**Computational Cost.** The cost of a single forward pass with hyper-dual numbers is of the same order of time and space complexity as the original function call. This reduces the memory cost compared to reverse-mode AD. The forward pass with hyper-dual numbers scales with the number

---

[2]Note, we have left our function definition as being a scalar output for pedagogical reasons but nothing precludes a vector or matrix output, which is required for the composition of functions in most ML architectures.

[3]During review, it was highlighted by an insightful reviewer that an implementation of nested forward-mode AD in JAX (Bradbury et al., 2018) would improve the potential adoption of the method. The reviewer even provided code, where the implementation is now included in §D.

[4]While interesting, these results might not immediately seem attractive to the ML community. The Hessian calculation of a model with parameters, $\boldsymbol{\theta} \in \mathbb{R}^D$, would require $D \cdot (D + 1)/2$ function evaluations, whereas Hessian vector products are widely available in many AD libraries leveraging tricks such as a forward over reverse routine (Paszke et al., 2019; Bradbury et al., 2018).

of parameters in the same way as the original function call. However, there is a constant overhead (no change in computational complexity) to be paid in the form of evaluating the second order terms throughout a forward pass. An example is that the addition of two scalars now requires 4 additions, and multiplication now requires 9 products and 5 additions. However, unlike for reverse-mode, these intermediate values can be overwritten once they have been used to propagate gradient information.

## 5 FORWARD-MODE OPTIMIZATION WITH SECOND ORDER INFORMATION

We now introduce FoMoH-$K$D, our new Forward-Mode Second-Order Hyperplane Search. To perform FoMoH-$K$D, we sample $K$ tangent vectors to build a $K$-dimensional hyperplane. We then perform a search in the sampled hyperplane to result in the new update step. We first introduce FoMoH-$K$D for the $K = 2$ example before generalizing to any $K$. Therefore, starting with $K = 2$, we take two search directions, $\mathbf{v}_1$ and $\mathbf{v}_2$, and we build a $2 \times 2$ matrix to form a Hessian in the affine hyperplane defined by $\boldsymbol{\theta} + \kappa_1 \mathbf{v}_1 + \kappa_2 \mathbf{v}_2$. We evaluate a function, $f(\cdot)$, with a hyper-dual number using the pairs $\{\mathbf{v}_1, \mathbf{v}_1\}$, $\{\mathbf{v}_1, \mathbf{v}_2\}$, and $\{\mathbf{v}_2, \mathbf{v}_2\}$ for the $\epsilon_1, \epsilon_2$ coefficients. The result is the Hessian, $\tilde{\mathbf{H}}_{2 \times 2}$, in the $2 \times 2$ plane, and corresponding step sizes, $\kappa_1$ and $\kappa_2$, to take in each search direction:

$$\tilde{\mathbf{H}}_{2 \times 2} = \left[ \begin{array}{cc} \mathbf{v}_1^\top \nabla^2 f(\boldsymbol{\theta}) \mathbf{v}_1 & \mathbf{v}_1^\top \nabla^2 f(\boldsymbol{\theta}) \mathbf{v}_2 \\ \mathbf{v}_2^\top \nabla^2 f(\boldsymbol{\theta}) \mathbf{v}_1 & \mathbf{v}_2^\top \nabla^2 f(\boldsymbol{\theta}) \mathbf{v}_2 \end{array} \right], \qquad \left[ \begin{array}{c} \kappa_1 \\ \kappa_2 \end{array} \right] = -\tilde{\mathbf{H}}_{2 \times 2}^{-1} \left[ \begin{array}{c} \mathbf{v}_1^\top \nabla f(\boldsymbol{\theta}) \\ \mathbf{v}_2^\top \nabla f(\boldsymbol{\theta}) \end{array} \right] \quad (1)$$

As a result, we formulate a new update step (for minimization),

$$\boldsymbol{\theta}' = \boldsymbol{\theta} + \kappa_1 \mathbf{v}_1 + \kappa_2 \mathbf{v}_2.$$

We then extend the above result to any $K$-dimensional hyperplane by sampling $K$ search directions and evaluating the corresponding update step:

$$\boldsymbol{\theta}' = \boldsymbol{\theta} + \eta \sum_{k=1}^{K} \kappa_k \mathbf{v}_k. \quad (2)$$

This resulting generalized hyperplane update step allows one to trade-off computational cost with the size of the search space. For example, the cost of evaluating a $K$-dimensional Hessian and then invert it is $\mathcal{O}(K^3)$, which is feasible for small enough $K$.[5] Our new forward-mode hyperplane search, FoMoH-$K$D, opens up the possibility of transitioning between a line search, when $K = 1$, all the way to a full Newton step, when $K = D$, where we include a theorem in §5.1 (with proof in §B.1) and demonstrate empirically in §6. The pseudo-code for a single update step is given in Algorithm 1, with the overall FoMoH-$K$D routine provided in §A.

### 5.1 CONVERGENCE PROPERTIES OF FoMoH-$K$D

We now state the convergence properties of FoMoH-$K$D, where we provide detailed proofs in §B. This theorem relates the convergence of FoMoH-$K$D to the dimension of the hyperplane, $K$, while bounding according to the geometry of the function:

**Theorem 1.** *Let $f : \mathbb{R}^D \to \mathbb{R}$ be a strongly convex quadratic function with global minimizer $\boldsymbol{\theta}^*$ and Hessian $\mathbf{A}$. Then the norm of the expected error converges geometrically with bounds,*

$$\frac{D - K}{D} \left( \sqrt{\mathrm{cond}(\mathbf{A})} \right)^{-1} \|\mathbf{e}_t\|_2 \leq \|\mathbb{E}[\mathbf{e}_{t+1}]\|_2 \leq \frac{D - K}{D} \sqrt{\mathrm{cond}(\mathbf{A})} \|\mathbf{e}_t\|_2,$$

*that depend on the condition number of $\mathbf{A}$ where $\|\mathbf{e}_{t+1}\| = \|\boldsymbol{\theta}_{t+1} - \boldsymbol{\theta}^*\|$ and $K$ is the dimension of the hyperplane.*

To arrive at this theorem, we perform a variable transformation to result in a diagonal distance metric on the new function space. We then use the expectation properties the projection matrix, where the projection matrix is used calculate the minimum distance of the global minimizer to the sampled affine hyperplane. This measurement compares the errors between sequential steps of FoMoH-$K$D leading to the result of Theorem 1. Full details are included in §B.

As a direct consequence of Theorem 1, we have the following corollary:

---

[5]For instances where $\tilde{\mathbf{H}}_{K \times K}$ is not invertible, we add jitter to the diagonal. This seems to work well.

**Corollary 1.1.** *When $D = K$, the update step of FoMoH-KD follows Newton's method.*

This corollary is to be expected for a quadratic function if FoMoH-$K$D is to generalize to Newton's method, since Newton's step only requires a single step, hence $\|\mathbf{e}_{t+1}\| = 0$ when $D = K$.

Overall we can directly compare FoMoH-$K$D's convergence geometric convergence rate in expectation with the sub-linear convergence rate (in expectation) of FGD, which we also show in §B.4.[6] This result is significant and means that we expect FoMoH-$K$D to have a faster convergence rate than FGD. We consistently see this result in practice as shown in the following experiments section.

---

**Algorithm 1** FoMoH-$K$D, hyperplane update step for function, $f$, and parameters, $\boldsymbol{\theta} \in \mathbb{R}^D$.

---

**Define:** HyperPlaneStep$(f, \boldsymbol{\theta}, K)$
  **Set:** $N = (K^2 + K)/2$, $\boldsymbol{\Theta} \in \mathbb{R}^{N \times D}$, $\boldsymbol{V} \in \mathbb{R}^{K \times D}$, $\boldsymbol{V}_1 \in \mathbb{R}^{N \times D}$, $\boldsymbol{V}_2 \in \mathbb{R}^{N \times D}$
  # For loop vectorized in code.
  **for** $n = 1, \ldots, N$ **do**
    $\boldsymbol{\Theta}[n, :] = \boldsymbol{\theta}$     # Repeat current parameter values for vectorized evaluation.
  **end for**
  **for** $k = 1, \ldots, K$ **do**
    $\boldsymbol{V}[k, :] = \mathbf{v} \sim \mathcal{N}(\mathbf{0}, \mathbf{I})$     # Sample $K$ tangent vectors to build hyperplane.
  **end for**
  # Vectorize tangent vectors for evaluation of all $N$ elements of $\tilde{\mathbf{H}}_{K \times K}$.
  $l = 1$
  **for** $i = 1, \ldots, K$ **do**
    **for** $j = i, \ldots, K$ **do**
      $\boldsymbol{V}_1[l, :] = \boldsymbol{V}[i, :]$
      $\boldsymbol{V}_2[l, :] = \boldsymbol{V}[j, :]$
      $l = l + 1$
    **end for**
  **end for**
  $\mathbf{z}_0 + \mathbf{z}_1 \epsilon_1 + \mathbf{z}_2 \epsilon_2 + \mathbf{z}_{12} \epsilon_1 \epsilon_2 = f(\boldsymbol{\Theta} + \boldsymbol{V}_1 \epsilon_1 + \boldsymbol{V}_2 \epsilon_2 + \mathbf{0} \epsilon_1 \epsilon_2)$
  # Build $\tilde{\mathbf{H}}_{K \times K}$ and directional derivatives vector, $\tilde{\mathbf{G}}_K$, in order to evaluate equation 2.
  **Set:** $\tilde{\mathbf{H}}_{K \times K} \in \mathbb{R}^{K \times K}$, $\tilde{\mathbf{G}}_K \in \mathbb{R}^{K \times 1}$
  $l = 1$
  **for** $i = 1, \ldots, K$ **do**
    $\tilde{\mathbf{G}}_K[i] = \mathbf{z}_1[l]$
    **for** $j = i, \ldots, K$ **do**
      $\tilde{\mathbf{H}}_{K \times K}[i, j] = \mathbf{z}_{12}[k]$
      $\tilde{\mathbf{H}}_{K \times K}[j, i] = \mathbf{z}_{12}[k]$
      $l = l + 1$
    **end for**
  **end for**
  # Returns update direction of vector size $D$
  **return** $\sum_k ((-\tilde{\mathbf{H}}_{K \times K}^{-1} \tilde{\mathbf{G}}_K)[k] \cdot \boldsymbol{V}[k, :])$

---

## 6 EXPERIMENTS

In the following experiments section, we investigate how the performance of FoMoH-$K$D compares to FGD. Performance is measured by the speed of convergence and the resulting final reported loss of an optimizer. We summarize the key experimental results before describing them in detail:

- We first use the Rosenbrock function to show how FoMoH-$K$D's convergence rate increases with $K$ until it reaches the same performance of Newton's method (thus verifying the theory empirically).

- We also use the Rosenbrock function to show the sensitivity of FGD to its learning rate, whereas FoMoH-$K$D does not suffer from this with its learning rate fixed at $1.0$.

---

[6]We also derive the variance in FGD's estimator in §B.3.

- We then demonstrate FoMoH-$K$D's faster convergence rate across the two learning tasks of logistic regression and CNN classification. We use these two problems to investigate the performance of FoMoH-$K$D compared to FGD as we increase from 7,850 parameters to 431,080 parameters.

Across the experiments, we also include BP (Stochastic Gradient Descent) performance for reference but we emphasize that our focus is on theoretically and empirically improving the forward-mode AD as demonstrated by the significant improvement of FoMoH-$K$D over FGD.

## 6.1 ROSENBROCK FUNCTION

The Rosenbrock function (Rosenbrock, 1960), $f(\boldsymbol{\theta}) = \sum_{i=1}^{D-1}(100(\theta_{i+1} - \theta_i^2)^2 + (1 - \theta_i)^2)$, is designed to be a challenging test case for non-convex optimization where the solution, $f(\mathbf{1}) = 0$, falls inside a narrow valley. The learning rate for the FoMoH-$K$D variants is set to 1.0 for the Rosenbrock experiments. FoMoH-BP is defined as FoMoH-1D but using reverse-mode AD to evaluate the gradient as the single tangent vector (see §C.1 for details).

**Robustness of FoMoH-$K$D for 2D Rosenbrock.** We illustrate the variance in a single update step starting from a randomly sampled point in Figure 1a for the 2D Rosenbrock function. We compare FGD, FoMoH-1D, FoMoH-BP, FoMoH-2D, and Newton's method $(-(\nabla^2 f(\boldsymbol{\theta}))^{-1} \nabla f(\boldsymbol{\theta}))$. We plot the expected (average) step across 10,000 samples for all approaches shown via the plotted lines. We also plot the distribution of sampled steps by superimposing a scatter plot in the corresponding approach's color. For FGD, we see that the expected descent direction is the same as the gradient at that point, hence the alignment with FoMoH-BP that directly calculates the gradient. This plot highlights the reliance on a well-chosen learning rate for FGD. When the gradient is known, like for reverse-mode approaches such as FoMoH-BP, the step size is automatically normalized by the local curvature along the gradient (purple). Moving to our proposed approach of FoMoH-$K$D, FoMoH-1D (in green) has a descent direction that differs from the gradient and is governed by the distribution of samples that fall on the ellipse defined by the Hessian, $\nabla^2 f(\boldsymbol{\theta})$. Since $D = 2$ for this example, FoMoH-2D takes the step on this ellipse that picks the local minimum of the quadratic approximation, shown in red. This is also the same update produced by Newton's step (black dashed lines), where Figure 1a shows the two are directly aligned, following Corollary 1.1.

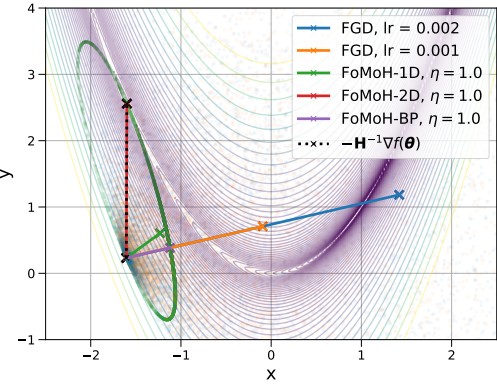
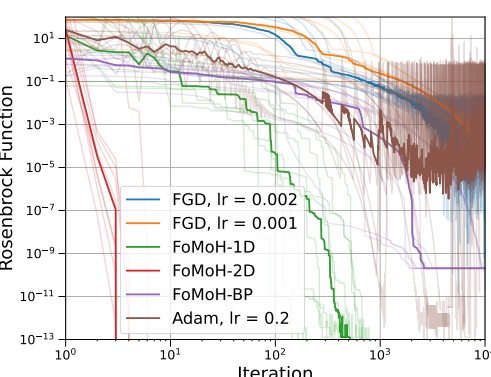

(a) Expected step taken by the stochastic approaches of FoMoH-$K$D and FGD. Included is a single Newton step for reference. For FoMoH-$K$D, the samples show how the curvature constrains the step size, compared to the sensitivity of FGD to the learning rate. For this 2D function, FoMoH-2D and Newton's step are aligned.

(b) Comparison of the stochastic approaches in minimization of the 2D Rosenbrock function. Average performance (median) is shown over 10 random initial conditions. FoMoH-1D outperforms all first-order approaches, with FoMoH-2D converging in orders of magnitude faster than all methods.

Figure 1: Results over the 2D Rosenbrock function.

**Faster Convergence rate of FoMoH-$K$D for 2D Rosenbrock.** In Figure 1b, we compare the performance of different optimization routines for the 2D Rosenbrock function. These methods use the same 10 randomly sampled starting locations. We highlight the median performing run with the thicker line. Both axes are on the log scale and show the significant advantage of FoMoH-$K$D, with $K = 2$. In this illustrative example, we see the faster convergence rate and better overall optimization performance of FoMoH-$K$D (including over BP).

**Increasing $K$ for the 10D Rosenbrock Function.** We now evaluate the performance of FoMoH-$K$D as we increase $K$ from 2 to the input dimension of the function. Figure 2 shows this comparison for the 10D Rosenbrock function, where we use 10 random initializations for the different $K$. The median performance for each $K$ is then shown, where we see a perfect ordering of performance that aligns with the dimension of the hyperplane. The best performing FoMoH-$K$D is for $K = D$, with the worst corresponding to the lowest dimension implemented, $K = 2$. This demonstrates Corollary 1.1 in practice, where FoMoH-$K$D trends towards Newton's method as $K$ tends to $D$, where we actually see the median performances of FoMoH-10D and Newton's method aligned.

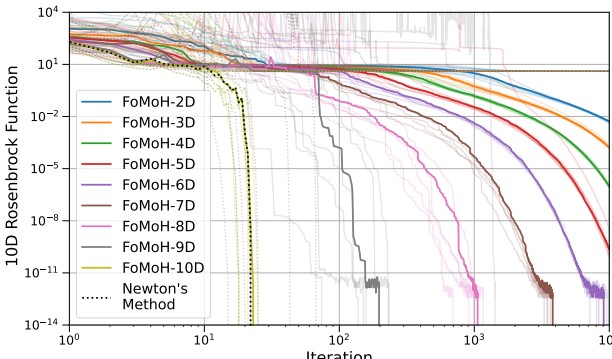

Figure 2: Performance of FoMoH-$K$D for $K = 2 \dots 10$ on the 10D Rosenbrock function. Solid lines represent the median, with transparent lines corresponding to the each of the 10 random seeds. There is a clear pattern of higher dimensions performing better, with the performance of $K = 10$ coinciding with Newton's Method (black dotted line). For an equivalent figure with wall-clock time on the x-axis, see §E.

## 6.2 MULTINOMIAL LOGISTIC REGRESSION

We now compare the performance of FoMoH-$K$D for a multinomial logistic regression model with 7,850 parameters applied to the MNIST dataset (LeCun et al., 1998a). Table 1 displays mean and standard deviation performance for each approach, where the forward-mode-only approaches are highlighted separately from the methods that include reverse-mode steps. Additional details on hyperparameter optimization are included in Appendix C.3. Figure 3 displays the training and validation curves for both accuracy and negative log-likelihood (NLL) for the forward-mode approaches (Figure 7 in appendix includes additional reverse-mode approaches). We see an improvement in **speed of convergence** as $K$ increases. We also see that all FoMoH-$K$D variants, with a fixed learning rate, degrade in performance after reaching their local minimum (NLL) or maximum (accuracy). Therefore the introduction of a **learning rate scheduler** mitigates this behavior, where we reduce the learning rate by a factor after a given number of epochs (see §C.3). Here, FGD is competitive with the FoMoH-$K$D variants but is slower to converge. In §6.3 we show how FGD degrades in performance for a larger parameter space.

## 6.3 CONVOLUTIONAL NEURAL NETWORK

We now demonstrate how FoMoH-$K$D compares to FGD for the larger dimensional learning problem of a convolutional neural network (CNN) with 431,080 parameters applied to the MNIST dataset. Both Table 2 and Figure 4 highlight the advantage of FoMoH-$K$D over FGD. **For the larger parameter space FGD requires more epochs to converge compared to all FoMoH-$K$D variants.** The learning rate scheduler, which decays the learning rate by a factor of 10 every 1,000 epochs, further improves FoMoH-$K$D by helping to avoid getting stuck in low performance regions. Here, we see the clear trend that the best performing forward-mode-only approach comes from the largest $K$, which was $K = 3$ for this experiment. We compare to BP-based approaches to provide context to the reader, but highlight that we do not expect forward-mode-only approaches to outperform well-engineered BP methods yet as noted in literature (Baydin et al., 2022; Ren et al., 2022; Silver et al., 2021; Fournier et al., 2023). Instead, we have shown that there is significant potential

Table 1: Multinomial logistic regression results for MNIST. When comparing the forward-mode-only approaches in the upper section of the table, we see improvement in performance with increasing hyperplane dimension for FoMoH-$K$D. For this multinomial logistic regression example, FoMoH-3D and FoMoH-2D with learning rate schedulers, are competitive with FGD. However we will see this result change with a larger dimensional problem in §6.3. Both reverse-mode approaches in the lower section of the table have similar performance, and are included for reference.

| APPROACH | TRAINING LOSS | VALIDATION LOSS | TRAINING ACCURACY | VALIDATION ACCURACY |
|---|---|---|---|---|
| FGD | $0.2976 \pm 0.0007$ | $\mathbf{0.2949 \pm 0.0017}$ | $0.9154 \pm 0.0003$ | $0.9163 \pm 0.0019$ |
| FoMoH-1D | $0.3223 \pm 0.0013$ | $0.3186 \pm 0.0019$ | $0.9073 \pm 0.0011$ | $0.9110 \pm 0.0021$ |
| FoMoH-1D (LR-Sch.) | $0.3192 \pm 0.0012$ | $0.3160 \pm 0.0025$ | $0.9085 \pm 0.0011$ | $0.9118 \pm 0.0021$ |
| FoMoH-2D | $0.3010 \pm 0.0018$ | $0.3015 \pm 0.0031$ | $0.9144 \pm 0.0009$ | $0.9149 \pm 0.0015$ |
| FoMoH-2D (LR-Sch.) | $0.2921 \pm 0.0014$ | $0.2951 \pm 0.0027$ | $0.9174 \pm 0.0005$ | $\mathbf{0.9170 \pm 0.0010}$ |
| FoMoH-3D | $0.3449 \pm 0.0017$ | $0.3343 \pm 0.0034$ | $0.8999 \pm 0.0008$ | $0.9036 \pm 0.0023$ |
| FoMoH-3D (LR-Sch.) | $\mathbf{0.2893 \pm 0.0017}$ | $0.3054 \pm 0.0034$ | $\mathbf{0.9195 \pm 0.0007}$ | $0.9153 \pm 0.0010$ |
| FoMoH-BP | $0.2312 \pm 0.0001$ | $0.2679 \pm 0.0003$ | $0.9366 \pm 0.0001$ | $0.9267 \pm 0.0002$ |
| BACKPROPAGATION | $0.2265 \pm 0.0000$ | $0.2710 \pm 0.0001$ | $0.9381 \pm 0.0001$ | $0.9267 \pm 0.0003$ |

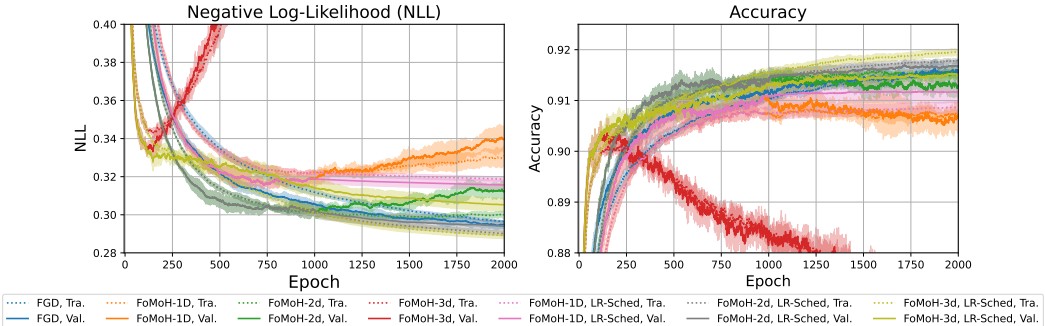

Figure 3: Forward-mode training and validation curves for the multinomial logistic regression model on the MNIST dataset. Average and standard deviation is shown for five random initializations. Increasing $K$, in combination with a learning rate scheduler to regularize steps, increases performance in terms of a lower NLL and a higher accuracy for FoMoH-$K$D.

in using second-order information to scale the performance of forward-mode optimization to larger dimensions and how **our approach outperforms the only previous forward-mode-only approach of FGD**. Furthermore, we have shown that increasing $K$ for FoMoH-$K$D improves optimization performance. For additional details on hyperparameter selection using Bayesian optimization, please refer to Appendix C.4.

Table 2: CNN results for MNIST. The forward-mode-only approaches in the upper section of the table show that FoMoH-$K$D's performance improves with the dimension of $K$, especially when used with the learning rate scheduler. FoMoH-3D outperforms FoMoH-2D, FoMoH-1D, and FGD. The reverse-mode approaches (included only for reference) in the lower section outperform forward-mode, with BP slightly better than FoMoH-BP.

| APPROACH | TRAINING LOSS | VALIDATION LOSS | TRAINING ACCURACY | VALIDATION ACCURACY |
|---|---|---|---|---|
| FGD | $0.1211 \pm 0.0097$ | $0.1104 \pm 0.0086$ | $0.9641 \pm 0.0038$ | $0.9677 \pm 0.0030$ |
| FoMoH-1D | $0.1663 \pm 0.0069$ | $0.1571 \pm 0.0092$ | $0.9518 \pm 0.0011$ | $0.9550 \pm 0.0002$ |
| FoMoH-1D (LR-Sch.) | $0.1617 \pm 0.0119$ | $0.1575 \pm 0.0158$ | $0.9515 \pm 0.0035$ | $0.9539 \pm 0.0034$ |
| FoMoH-2D | $0.1015 \pm 0.0041$ | $0.1016 \pm 0.0066$ | $0.9691 \pm 0.0011$ | $0.9693 \pm 0.0020$ |
| FoMoH-2D (LR-Sch.) | $0.0900 \pm 0.0070$ | $0.0913 \pm 0.0041$ | $0.9731 \pm 0.0022$ | $0.9718 \pm 0.0014$ |
| FoMoH-3D | $0.1085 \pm 0.0105$ | $0.1073 \pm 0.0133$ | $0.9674 \pm 0.0022$ | $0.9686 \pm 0.0028$ |
| FoMoH-3D (LR-Sch.) | $\mathbf{0.0809 \pm 0.0061}$ | $\mathbf{0.0923 \pm 0.0132}$ | $\mathbf{0.9759 \pm 0.0014}$ | $\mathbf{0.9734 \pm 0.0022}$ |
| FoMoH-BP | $0.0093 \pm 0.0016$ | $0.0310 \pm 0.0006$ | $0.9981 \pm 0.0005$ | $0.9903 \pm 0.0003$ |
| BACKPROPAGATION | $0.0053 \pm 0.0034$ | $0.0329 \pm 0.0032$ | $0.9990 \pm 0.0009$ | $0.9909 \pm 0.0004$ |

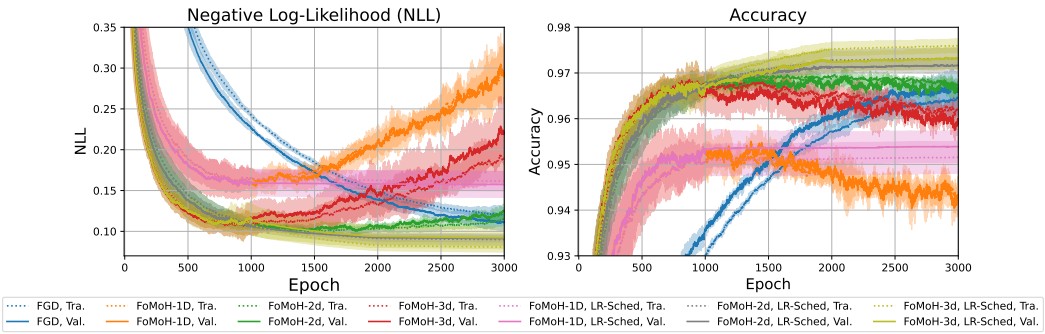

Figure 4: Forward-mode training and validation curves for the CNN on the MNIST dataset. Average and standard deviation is shown for three random initializations. Note how FGD (blue) is much slower to converge, with FoMoH-$K$D improving in performance with increasing $K$.

## 7 CONCLUSION

In this paper, we introduced FoMoH-$K$D, a novel forward-mode-only hyperplane search that uses second-order information. We provide theory to go along with our method, where we show, under certain assumptions, that FoMoH-$K$D generalizes to Newton's method for $K = D$, and also that the expected reduction in error from the minimum decays at an exponential rate of $(D - K)/D$. This compares with the previous approach of FGD, which has a sub-linear convergence rate as we show in §B.4. In addition to the proposed new approach and theory, we also demonstrated FoMoH-$K$D across multiple problem settings, where we were able to empirically support the theory. For the Rosenbrock function, we directly showed how FoMoH-$K$D tends to Newton's method, which is consistent with Corollary 1.1. This significant result is shown in Figure 2. For the learning tasks of multinomial logistic regression and CNN classification, we see how the first-order optimization approach of FGD degrades with increasing dimension of the parameter space. We do not see this degradation for FoMoH-$K$D, and we also observe that the second-order information means fewer epochs are needed to reach a better performance. This has the broader impact of improving efficiency, reducing cost, and increasing accuracy in ML optimization routines. As a further contribution, we provide a Python package that implements the AD backend and interfaces with PyTorch.[7]

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

SUPPLEMENTARY MATERIALS

## A FoMoH-$K$D ALGORITHM

---
**Algorithm 2** FoMoH-$K$D

---
**Require:** Objective function $f(\boldsymbol{\theta})$, initial point $\boldsymbol{\theta}$, step size $\eta$, hyperplane dimension $K$.
    **for** $t = 0, 1, 2, \ldots$ until convergence **do**
       $\mathbf{d} = \text{HyperPlaneStep}(f, \boldsymbol{\theta}, K)$ # See Algorithm 1
       $\boldsymbol{\theta} = \boldsymbol{\theta} + \eta\mathbf{d}$
    **end for**

---

We include a pictorial representation of the FoMoH-$K$D update step in Figure 5 for the 2D Rosenbrock function. This shows how equation 1 and equation 2 are applied.

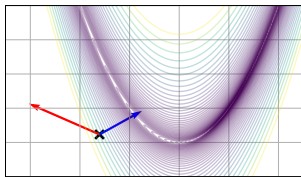 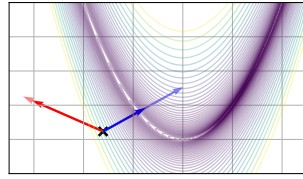 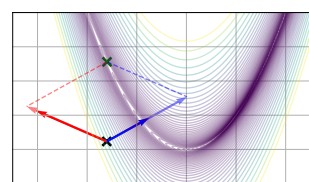

(a) Sample $K = 2$ tangent vectors: $\mathbf{v}_1, \mathbf{v}_2 \sim \mathcal{N}(\mathbf{0}, \mathbf{I})$.

(b) Apply equation 1 to project along the tangent vectors:
$$\begin{bmatrix} \kappa_1 \\ \kappa_2 \end{bmatrix} = -\tilde{\mathbf{H}}_{2\times2}^{-1} \begin{bmatrix} \mathbf{v}_1^\top \nabla f(\boldsymbol{\theta}) \\ \mathbf{v}_2^\top \nabla f(\boldsymbol{\theta}) \end{bmatrix}$$

(c) Apply the update step to propose the new location: $\boldsymbol{\theta}' = \boldsymbol{\theta} + \kappa_1\mathbf{v}_1 + \kappa_2\mathbf{v}_2$

Figure 5: FoMoH-$K$D update step shown for $K = 2$ in 2D Rosenbrock function.

## B THEORY

The first two subsections are concerned with proving Theorem 1 for FoMoH-$K$D and the required property of the expectation of the projection matrix. The latter two are concerned with the variance and convergence properties of FGD.

### B.1 CONVERGENCE ANALYSIS OF FoMoH-$K$D

**Theorem 1.** *Let $f : \mathbb{R}^D \to \mathbb{R}$ be a strongly convex quadratic function with global minimizer $\boldsymbol{\theta}^*$ and Hessian $\mathbf{A}$. Then the norm of the expected error converges geometrically with bounds,*

$$\frac{D - K}{D} \left(\sqrt{\text{cond}(\mathbf{A})}\right)^{-1} \|\mathbf{e}_t\|_2 \leq \|\mathbb{E}[\mathbf{e}_{t+1}]\|_2 \leq \frac{D - K}{D} \sqrt{\text{cond}(\mathbf{A})} \|\mathbf{e}_t\|_2,$$

*that depend on the condition number of $\mathbf{A}$ where $\|\mathbf{e}_{t+1}\| = \|\boldsymbol{\theta}_{t+1} - \boldsymbol{\theta}^*\|$ and $K$ is the dimension of the hyperplane.*

*Proof.* Assuming a convex quadratic function $f : \mathbb{R}^D \to \mathbb{R}$, of form $f(\boldsymbol{\theta}) = \frac{1}{2}\boldsymbol{\theta}^\top \mathbf{A}\boldsymbol{\theta} + \boldsymbol{\theta}^\top \mathbf{b}$ and defining an affine subspace,

$$\mathcal{V}_\mathbf{x}^K = \left\{\mathbf{x} + \sum_{k=1}^K \kappa_k \mathbf{v}_k \,\middle|\, k \in \mathbb{N}, \mathbf{v}_k \in \mathbb{R}^D, \kappa \in \mathbb{R}, \mathbf{x} \in \mathbb{R}^D\right\},$$

we derive a new function $g(\boldsymbol{\kappa})$ for $\boldsymbol{\kappa} \in \mathbb{R}^K$ that represents $f(\boldsymbol{\theta})$ for when $\boldsymbol{\theta} \in \mathcal{V}_\mathbf{x}^K$ ($\boldsymbol{\theta}$ restricted to the hyperplane[8]). As a result, the linear map is given by $\boldsymbol{\theta} = \mathbf{V}\boldsymbol{\kappa}$, where $\mathbf{V} \in \mathbb{R}^{D\times K}$. Noting that the span is sampled according to an isotropic Gaussian distribution, $\mathbf{v}_i \sim \mathcal{N}(\mathbf{0}, \mathbf{I}_D)$.

---
[8]E.g., for $K = 2$, $g(\kappa_1, \kappa_2) = f(\mathbf{x} + \kappa_1\mathbf{v}_1 + \kappa_2\mathbf{v}_2)$

We explicitly write the FoMoH-$K$D update step from equation 1 as:

$$\boldsymbol{\kappa}_{t+1} = \boldsymbol{\kappa}_t - \left([\mathbf{v}_1, \mathbf{v}_2, ..., \mathbf{v}_K]^\top \nabla^2 f(\boldsymbol{\theta}_t)[\mathbf{v}_1, ..., \mathbf{v}_K]\right)^{-1} [\mathbf{v}_1 \cdot \nabla f(\boldsymbol{\theta}_t), ..., \mathbf{v}_K \cdot \nabla f(\boldsymbol{\theta}_t)]^\top$$

$$= \boldsymbol{\kappa}_t - \left(\mathbf{V}^\top \nabla^2 f(\boldsymbol{\theta}_t)\mathbf{V}\right)^{-1} \mathbf{V}^\top \nabla f(\boldsymbol{\theta}_t),$$

which is equivalent to Newton's method in $\boldsymbol{\kappa}$:

$$\boldsymbol{\kappa}_{t+1} = \boldsymbol{\kappa}_t - \left(\nabla^2 g(\boldsymbol{\kappa}_t)\right)^{-1} \nabla g(\boldsymbol{\kappa}_t),$$

where each $k^{\text{th}}$ component of $\nabla g(\boldsymbol{\kappa}_t)$ corresponds to the derivative of $f(\boldsymbol{\theta}_t)$ along $\mathbf{v}_k$ by definition. Following equation 2, we use $\mathbf{V}$ to project back to $\boldsymbol{\theta}$:

$$\mathbf{V}\boldsymbol{\kappa}_{t+1} = \mathbf{V}\boldsymbol{\kappa}_t - \mathbf{V}\left(\mathbf{V}^\top \nabla^2 f(\boldsymbol{\theta}_t)\mathbf{V}\right)^{-1} \mathbf{V}^\top \nabla f(\boldsymbol{\theta}_t),$$

$$\boldsymbol{\theta}_{t+1} = \boldsymbol{\theta}_t - \mathbf{V}\left(\mathbf{V}^\top \nabla^2 f(\boldsymbol{\theta}_t)\mathbf{V}\right)^{-1} \mathbf{V}^\top \nabla f(\boldsymbol{\theta}_t). \tag{3}$$

After defining the FoMoH-$K$D update step, we now perform a parameter transformation to simplify the steps thereafter. As such, since $\mathbf{A}$ is positive definite, we diagonalize $\mathbf{A}$ using an eigenvalue decomposition, $\mathbf{A} = \mathbf{G}\boldsymbol{\Lambda}\mathbf{G}^\top$. Therefore, without loss of generality, we transform our variables to $\tilde{\boldsymbol{\theta}} = \boldsymbol{\Lambda}^{1/2}\mathbf{G}^\top \boldsymbol{\theta}$. This gives $f(\tilde{\boldsymbol{\theta}}) = \frac{1}{2}\tilde{\boldsymbol{\theta}}^\top \tilde{\boldsymbol{\theta}} + \tilde{\boldsymbol{\theta}}^\top \boldsymbol{\Lambda}^{-1/2}\mathbf{G}^\top \mathbf{b}$.

We now show that the FoMoH-$K$D update step in this transformed space, $\tilde{\boldsymbol{\theta}}$, is equivalent to finding to the closest point on the affine hyperplane to $\tilde{\boldsymbol{\theta}}^*$. We will use the definition of the projection matrix, $\mathbf{P} = \mathbf{V}(\mathbf{V}^\top \mathbf{V})^{-1}\mathbf{V}^\top$, which maps any point $\tilde{\boldsymbol{\theta}}$ to its closest point on the hyperplane defined by $\mathbf{V}$. We start from the FoMoH-$K$D update step applied to the transformed space $\tilde{\boldsymbol{\theta}}$:

$$\tilde{\boldsymbol{\theta}}_{t+1} = \tilde{\boldsymbol{\theta}}_t - \mathbf{V}\left(\mathbf{V}^\top \nabla^2 f(\tilde{\boldsymbol{\theta}}_t)\mathbf{V}\right)^{-1} \mathbf{V}^\top \nabla f(\tilde{\boldsymbol{\theta}}_t),$$

and then using $\nabla^2 f(\tilde{\boldsymbol{\theta}}_t) = \mathbf{I}$, and $\nabla f(\tilde{\boldsymbol{\theta}}_t) = \tilde{\boldsymbol{\theta}}_t + \boldsymbol{\Lambda}^{-1/2}\mathbf{G}^\top \mathbf{b}$ gives:

$$\tilde{\boldsymbol{\theta}}_{t+1} = \tilde{\boldsymbol{\theta}}_t - \mathbf{V}\left(\mathbf{V}^\top \mathbf{V}\right)^{-1} \mathbf{V}^\top \left(\tilde{\boldsymbol{\theta}}_t + \boldsymbol{\Lambda}^{-1/2}\mathbf{G}^\top \mathbf{b}\right),$$

Finally, noting that $\tilde{\boldsymbol{\theta}}^* = -\boldsymbol{\Lambda}^{-1/2}\mathbf{G}^\top \mathbf{b}$ from setting the gradient to zero, we get

$$\tilde{\boldsymbol{\theta}}_{t+1} = \tilde{\boldsymbol{\theta}}_t - \mathbf{V}\left(\mathbf{V}^\top \mathbf{V}\right)^{-1} \mathbf{V}^\top \left(\tilde{\boldsymbol{\theta}}_t - \tilde{\boldsymbol{\theta}}^*\right),$$

$$= \tilde{\boldsymbol{\theta}}_t + \mathbf{P}\left(\tilde{\boldsymbol{\theta}}^* - \tilde{\boldsymbol{\theta}}_t\right).$$

Therefore, we arrive at the same update rule as in equation 2 for FoMoH-$K$D that now relates the update rule to $\tilde{\boldsymbol{\theta}}^*$. We use the equivalence of this rule, $\tilde{\boldsymbol{\theta}}_{t+1} = \tilde{\boldsymbol{\theta}}_t + \mathbf{P}(\tilde{\boldsymbol{\theta}}^* - \tilde{\boldsymbol{\theta}}_t)$, to show how the convergence rate of FoMoH-$K$D relates to $K$ and $D$. We use the definition of the expectation of the projection matrix $\mathbb{E}[\mathbf{P}] = \frac{K}{D}\mathbf{I}$ (see §B.2 for proof), such that

$$\mathbb{E}[\tilde{\boldsymbol{\theta}}_{t+1}] = \mathbb{E}[\tilde{\boldsymbol{\theta}}_t + \mathbf{P}(\tilde{\boldsymbol{\theta}}^* - \tilde{\boldsymbol{\theta}}_t)]$$

$$= \tilde{\boldsymbol{\theta}}_t + \frac{K}{D}(\tilde{\boldsymbol{\theta}}^* - \tilde{\boldsymbol{\theta}}_t). \tag{4}$$

This key result is consistent with our observation that FoMoH-$K$D generalizes to Newton's method, as when $K = D$ equation 4 reduces to $\tilde{\boldsymbol{\theta}}_{t+1} = \tilde{\boldsymbol{\theta}}^*$, and therefore $\boldsymbol{\theta}_{t+1} = \boldsymbol{\theta}^*$ (since $\tilde{\boldsymbol{\theta}}^* = \boldsymbol{\Lambda}^{1/2}\mathbf{G}^\top \boldsymbol{\theta} = -\boldsymbol{\Lambda}^{-1/2}\mathbf{G}^\top \mathbf{b} \implies \boldsymbol{\theta} = -\mathbf{A}^{-1}\mathbf{b} = \boldsymbol{\theta}^*$). Furthermore, we remove $\tilde{\boldsymbol{\theta}}^*$ from both sides of the equation and transform back to the original parameter space:

$$\mathbb{E}[\tilde{\boldsymbol{\theta}}_{t+1} - \tilde{\boldsymbol{\theta}}^*] = \tilde{\boldsymbol{\theta}}_t + \frac{K}{D}(\tilde{\boldsymbol{\theta}}^* - \tilde{\boldsymbol{\theta}}_t) - \tilde{\boldsymbol{\theta}}^*$$

$$\|\mathbb{E}[\tilde{\boldsymbol{\theta}}_{t+1} - \tilde{\boldsymbol{\theta}}^*]\| = \frac{D - K}{D}\|\tilde{\boldsymbol{\theta}}_t - \tilde{\boldsymbol{\theta}}^*\|$$

$$\|\mathbb{E}[\boldsymbol{\Lambda}^{1/2}\mathbf{G}^\top \boldsymbol{\theta}_{t+1} - \boldsymbol{\Lambda}^{1/2}\mathbf{G}^\top \boldsymbol{\theta}^*]\| = \frac{D - K}{D}\|\boldsymbol{\Lambda}^{1/2}\mathbf{G}^\top \boldsymbol{\theta}_t - \boldsymbol{\Lambda}^{1/2}\mathbf{G}^\top \boldsymbol{\theta}^*\|$$

$$\|\mathbb{E}[\boldsymbol{\Lambda}^{1/2}\mathbf{G}^\top (\boldsymbol{\theta}_{t+1} - \boldsymbol{\theta}^*)]\| = \frac{D - K}{D}\|\boldsymbol{\Lambda}^{1/2}\mathbf{G}^\top (\boldsymbol{\theta}_t - \boldsymbol{\theta}^*)\|$$

$$\|\mathbb{E}[\mathbf{e}_{t+1}]\|_{\mathbf{A}} = \frac{D - K}{D}\|\mathbf{e}_t\|_{\mathbf{A}} \tag{5}$$

This implies that the Euclidean norm of the expected error converges geometrically, but with bounds that depend on the eigenvalues of $\mathbf{A}$. Using the relation $\sqrt{\lambda_{\min}}\|\mathbf{x}\|_2 \leq \|\mathbf{x}\|_{\mathbf{A}} \leq \sqrt{\lambda_{\max}}\|\mathbf{x}\|_2$, where $\lambda_{\min}, \lambda_{\max}$ are the maximum and minimum Eigenvalues of $\mathbf{A}$, we have the following bounds on the norm of the errors, $\mathbf{e}_t$, and $\mathbb{E}[\mathbf{e}_{t+1}]$:

$$\frac{D-K}{D}\sqrt{\lambda_{\min}(\mathbf{A})}\|\mathbf{e}_t\|_2 \leq \frac{D-K}{D}\|\mathbf{e}_t\|_{\mathbf{A}} \leq \frac{D-K}{D}\sqrt{\lambda_{\max}(\mathbf{A})}\|\mathbf{e}_t\|_2,$$

$$\sqrt{\lambda_{\min}(\mathbf{A})}\|\mathbb{E}[\mathbf{e}_{t+1}]\|_2 \leq \|\mathbb{E}[\mathbf{e}_{t+1}]\|_{\mathbf{A}} \leq \sqrt{\lambda_{\max}(\mathbf{A})}\|\mathbb{E}[\mathbf{e}_{t+1}]\|_2.$$

We use Eq. equation 5 to rearrange the above inequalities to bound the Euclidean norm of the update step, $\|\mathbb{E}[\mathbf{e}_{t+1}]\|_2$, in terms of the current error norm, giving an upper bound:

$$\sqrt{\lambda_{\min}(\mathbf{A})}\|\mathbb{E}[\mathbf{e}_{t+1}]\|_2 \leq \frac{D-K}{D}\sqrt{\lambda_{\max}(\mathbf{A})}\|\mathbf{e}_t\|_2,$$

$$\|\mathbb{E}[\mathbf{e}_{t+1}]\|_2 \leq \frac{D-K}{D}\frac{\sqrt{\lambda_{\max}(\mathbf{A})}}{\sqrt{\lambda_{\min}(\mathbf{A})}}\|\mathbf{e}_t\|_2,$$

and a lower bound:

$$\frac{D-K}{D}\sqrt{\lambda_{\min}(\mathbf{A})}\|\mathbf{e}_t\|_2 \leq \sqrt{\lambda_{\max}(\mathbf{A})}\|\mathbb{E}[\mathbf{e}_{t+1}]\|_2,$$

$$\frac{D-K}{D}\frac{\sqrt{\lambda_{\min}(\mathbf{A})}}{\sqrt{\lambda_{\max}(\mathbf{A})}}\|\mathbf{e}_t\|_2 \leq \|\mathbb{E}[\mathbf{e}_{t+1}]\|_2,$$

leading to the final bounds on the Euclidean norm of the expected error at $t+1$, where the condition number, $\text{cond}(\mathbf{A}) = {\lambda_{\max}(\mathbf{A})}/{\lambda_{\min}(\mathbf{A})}$:

$$\frac{D-K}{D}\left(\sqrt{\text{cond}(\mathbf{A})}\right)^{-1}\|\mathbf{e}_t\|_2 \leq \|\mathbb{E}[\mathbf{e}_{t+1}]\|_2 \leq \frac{D-K}{D}\left(\sqrt{\text{cond}(\mathbf{A})}\right)\|\mathbf{e}_t\|_2 \qquad (6)$$

$$\square$$

This result shows that the update step reduces the error by the fraction $0 \leq \frac{D-K}{D} < 1$, with a linear decrease in the rate of reduction in the error as $K$ increases, and with bounds that depend on the geometry of the function. Therefore FoMoH-$K$D's convergence is exponential in expectation with the rate $\frac{D-K}{D}$. This compares to the sub-linear convergence (in expectation) of FGD that we derive in §B.4.

## B.2 EXPECTATION OF PROJECTION MATRIX

In Theorem 1, we used the identity $\mathbb{E}[\mathbf{P}] = \frac{K}{D}\mathbf{I}_D$, where we subscript the identity matrix with its corresponding dimension. Here, we prove this identity.

**Theorem 2.** *Let $\mathbf{V} \in \mathbb{R}^{D \times K}$ be a matrix where each element, $\mathbf{V}_{ij} \sim \mathcal{N}(0,1)$. Then the expectation of the projection matrix, $\mathbf{P} = \mathbf{V}(\mathbf{V}^\top \mathbf{V})^{-1}\mathbf{V}^\top$, is given by $\mathbb{E}[\mathbf{P}] = \frac{K}{D}\mathbf{I}_D$.*

*Proof.* Let $\mathbf{R} \in \mathbb{R}^{D \times D}$ be an orthogonal matrix such that $\mathbf{R}\mathbf{R}^\top = \mathbf{I}_D$. Then rotating $\mathbf{V}$ gives $\mathbf{V}' = \mathbf{R}\mathbf{V}$. We then use the rotational invariance property of the normal distribution (see Prop. 3.3.2 of Vershynin (2018)) to give $\mathbf{V} \overset{d}{=} \mathbf{R}\mathbf{V}$, where $\overset{d}{=}$ denotes equality in distribution. We now define a transformed projection matrix:

$$\mathbf{P}' = \mathbf{V}'(\mathbf{V}'^\top \mathbf{V}')^{-1}\mathbf{V}'^\top = \mathbf{R}\mathbf{V}(\mathbf{V}^\top \mathbf{R}^\top \mathbf{R}\mathbf{V})^{-1}\mathbf{V}^\top \mathbf{R}^\top = \mathbf{R}\mathbf{P}\mathbf{R}^\top.$$

Since $\mathbf{V}' \overset{d}{=} \mathbf{V}$, then $\mathbf{P} \overset{d}{=} \mathbf{P}'$, and $\mathbb{E}[\mathbf{P}'] = \mathbb{E}[\mathbf{R}\mathbf{P}\mathbf{R}^\top] = \mathbf{R}\mathbb{E}[\mathbf{P}]\mathbf{R}^\top = \mathbb{E}[\mathbf{P}]$. As a result of the last equation, we get the relation $\mathbf{R}\mathbb{E}[\mathbf{P}] = \mathbb{E}[\mathbf{P}]\mathbf{R}$ by post-multiplying by $\mathbf{R}$. Therefore, $\mathbb{E}[\mathbf{P}]$ commutes with all $\mathbf{R}$ in the orthogonal group. Then using Schur's Lemma (e.g. as stated in §1.2 of Fulton & Harris (2013))

$$\mathbb{E}[\mathbf{P}] = c\mathbf{I}_D$$

for some constant $c$. To determine $c$, we take the trace of both sides, and use linearity of expectation to move the trace inside the expectation:

$$\text{tr}(\mathbb{E}[\mathbf{P}]) = \text{tr}(c\mathbf{I}_D)$$

$$\mathbb{E}[\text{tr}(\mathbf{V}(\mathbf{V}^\top\mathbf{V})^{-1}\mathbf{V}^\top)] = cD$$

Then, using the identity $\text{tr}(\mathbf{AB}) = \text{tr}(\mathbf{BA})$ for $\mathbf{A} \in \mathbb{R}^{D \times K}$ and $\mathbf{B} \in \mathbb{R}^{K \times D}$:

$$\mathbb{E}[\text{tr}(\mathbf{V}^\top\mathbf{V}(\mathbf{V}^\top\mathbf{V})^{-1})] = cD$$
$$\mathbb{E}[\text{tr}(I_K)] = cD$$
$$K = cD \implies c = K/D$$

Thus,

$$\mathbb{E}[\mathbf{P}] = \frac{K}{D}\mathbf{I}_D$$

$\square$

### B.3 VARIANCE ANALYSIS OF FORWARD GRADIENT DESCENT

If we start with the definition from Baydin et al. (2022) of the forward gradient:

$$g_i(\boldsymbol{\theta}) = \frac{\partial f}{\partial \theta_i}v_i^2 + \sum_{j \neq i}\frac{\partial f}{\partial \theta_j}v_i v_j,$$

The expectation is given by $\mathbb{E}[g_i(\boldsymbol{\theta})] = \frac{\partial f}{\partial \theta_i}$. It is of interest as to how the variance of this estimate behaves, i.e. $\text{Var}(g_i(\boldsymbol{\theta}))$. Using the definition, $\text{Var}(X) = \mathbb{E}[X^2] - \mathbb{E}^2[X]$, we can derive each component as:

$$\mathbb{E}^2[g_i(\boldsymbol{\theta})] = \left[\frac{\partial f}{\partial \theta_i}\right]^2,$$

and

$$\mathbb{E}[g_i(\boldsymbol{\theta})^2] = \mathbb{E}\left[\left[\frac{\partial f}{\partial \theta_i}\right]^2 v_i^4 + 2\frac{\partial f}{\partial \theta_i}\sum_{j \neq i}\frac{\partial f}{\partial \theta_j}v_i^3 v_j + \left(\sum_{j \neq i}\frac{\partial f}{\partial \theta_j}v_i v_j\right)^2\right].$$

We can expand the last term as:

$$\left(\sum_{j \neq i}\frac{\partial f}{\partial \theta_j}v_i v_j\right)^2 = v_i^2\left(\sum_{j \neq i}\left[\frac{\partial f}{\partial \theta_j}\right]^2 v_j^2 + 2\sum_{j \neq i}\sum_{k \neq i, k > j}\frac{\partial f}{\partial \theta_j}\frac{\partial f}{\partial \theta_k}v_j v_k\right),$$

then moving the expectation inside the square brackets and using the identities, $\mathbb{E}[X^2] = 1$, $\mathbb{E}[X^4] = 3$, $\mathbb{E}[X^3 Y] = 0$, $\mathbb{E}[X^2 Y^2] = 1$, and $\mathbb{E}[X^2 Y Z] = 0$ for the normal distribution gives:

$$\mathbb{E}[g_i(\boldsymbol{\theta})^2] = \left[\frac{\partial f}{\partial \theta_i}\right]^2 [3] + 2\frac{\partial f}{\partial \theta_i}\sum_{j \neq i}\frac{\partial f}{\partial \theta_j}[0] + \sum_{j \neq i}\left[\frac{\partial f}{\partial \theta_j}\right]^2 [1] + 2\sum_{j \neq i}\sum_{k \neq i, k > j}\frac{\partial f}{\partial \theta_j}\frac{\partial f}{\partial \theta_k}[0]$$

$$= 3\left[\frac{\partial f}{\partial \theta_i}\right]^2 + \sum_{j \neq i}\left[\frac{\partial f}{\partial \theta_j}\right]^2.$$

Finally, the variance is given by

$$\text{Var}(g_i(\boldsymbol{\theta})) = 2\left[\frac{\partial f}{\partial \theta_i}\right]^2 + \sum_{j \neq i}\left[\frac{\partial f}{\partial \theta_j}\right]^2.$$

### B.4 Convergence Analysis of FGD

Using the following assumptions:

- $f : \mathbb{R}^D \to \mathbb{R}$ is a convex function with minimum $f(\boldsymbol{\theta}^*)$.
- The update step (minimization) is: $\boldsymbol{\theta}_{t+1} = \boldsymbol{\theta}_t - \eta \mathbf{g}(\boldsymbol{\theta})$.
- $\mathbb{E}[\mathbf{g}(\boldsymbol{\theta})] = \nabla f(\boldsymbol{\theta})$, $\mathbb{E}[\mathbf{g}(\boldsymbol{\theta})^2] \leq G^2$ for some constant $G$. We derived this form above.
- $f$ has an $L$-Lipschitz continuous gradient: $\|\nabla f(\mathbf{x}) - \nabla f(\mathbf{y})\| \leq L\|\mathbf{x} - \mathbf{y}\|$.

We start with a first-order Taylor series expansion and input the update rule:

$$f(\boldsymbol{\theta}_{t+1}) \approx f(\boldsymbol{\theta}_t) + \nabla f(\boldsymbol{\theta}_t) \cdot (\boldsymbol{\theta}_{t+1} - \boldsymbol{\theta}_t) + \frac{1}{2}(\boldsymbol{\theta}_{t+1} - \boldsymbol{\theta}_t)^\top \nabla^2 f(\boldsymbol{\theta}_t)(\boldsymbol{\theta}_{t+1} - \boldsymbol{\theta}_t)$$

$$\approx f(\boldsymbol{\theta}_t) - \eta \nabla f(\boldsymbol{\theta}_t) \cdot \mathbf{g}(\boldsymbol{\theta}_t) + \frac{1}{2}\eta^2 \mathbf{g}(\boldsymbol{\theta}_t)^\top \nabla^2 f(\boldsymbol{\theta}_t)\mathbf{g}(\boldsymbol{\theta}_t).$$

Since the rate of change of the gradient is bounded due to the $L$-Lipschitz continuous gradient assumption, the curvature along any direction follows $\mathbf{g}(\boldsymbol{\theta}_t)^\top \nabla^2 f(\boldsymbol{\theta}_t)\mathbf{g}(\boldsymbol{\theta}_t) \leq L\|\mathbf{g}(\boldsymbol{\theta}_t)\|^2$. The next step is to combine this upper bound on the curvature with expectation operator to give the expected decrease in the function value:

$$\mathbb{E}[f(\boldsymbol{\theta}_{t+1})|\boldsymbol{\theta}_t] \leq \mathbb{E}\left[f(\boldsymbol{\theta}_t) - \eta \nabla f(\boldsymbol{\theta}_t) \cdot \mathbf{g}(\boldsymbol{\theta}_t) + \frac{L\eta^2}{2}\|\mathbf{g}(\boldsymbol{\theta}_t)\|^2 \Big| \boldsymbol{\theta}_t\right]$$

$$\mathbb{E}[f(\boldsymbol{\theta}_{t+1}) - f(\boldsymbol{\theta}_t)|\boldsymbol{\theta}_t] \leq -\eta\|\nabla f(\boldsymbol{\theta}_t)\|^2 + \frac{G^2 L\eta^2}{2}.$$

To infer the convergence rate, we want to see the rate in which the square of the gradient norm tends to zero since a magnitude of zero means the update rule has approaches a critical point, which will be the minimum, $f(\boldsymbol{\theta}^*)$, due to the convexity assumption. Following the outlined approach of Schmidt (2019), we usually want to know how many iterations we need to get to $\|\nabla f(\boldsymbol{\theta}_t)\|^2 \leq \epsilon$. Therefore we can rearrange the above to bound $\|\nabla f(\boldsymbol{\theta}_t)\|^2$ as

$$\eta\|\nabla f(\boldsymbol{\theta}_t)\|^2 \leq f(\boldsymbol{\theta}_t) - \mathbb{E}[f(\boldsymbol{\theta}_{t+1})|\boldsymbol{\theta}_t] + \frac{G^2 L\eta^2}{2}.$$

Introducing the summation over $T$ steps of the squared norms while using the fact that the first two terms on the right make up a telescoping series, such that $\sum_{t=1}^{T} \mathbb{E}[f(\boldsymbol{\theta}_t) - f(\boldsymbol{\theta}_{t+1})|\boldsymbol{\theta}_t] = \mathbb{E}[f(\boldsymbol{\theta}_1) - f(\boldsymbol{\theta}_{T+1})|\boldsymbol{\theta}_1]$, and assuming $f(\boldsymbol{\theta}_{T+1}) \geq f(\boldsymbol{\theta}^*)$, gives

$$\sum_{t=1}^{T} \eta\|\nabla f(\boldsymbol{\theta}_t)\|^2 \leq f(\boldsymbol{\theta}_1) - f(\boldsymbol{\theta}^*) + \sum_{t=1}^{T} \frac{G^2 L\eta^2}{2}.$$

Finally, we know that $\sum_{t=1}^{T} \|\nabla f(\boldsymbol{\theta}_t)\|^2 \geq \min_{t \in \{1,...,T\}} \{\|\nabla f(\boldsymbol{\theta}_t)\|^2\}$, giving

$$\min_{t \in \{1,...,T\}} \{\|\nabla f(\boldsymbol{\theta}_t)\|^2\} \sum_{t=1}^{T} \eta \leq f(\boldsymbol{\theta}_1) - f(\boldsymbol{\theta}^*) + \sum_{t=1}^{T} \frac{G^2 L\eta^2}{2}$$

$$\min_{t \in \{1,...,T\}} \{\|\nabla f(\boldsymbol{\theta}_t)\|^2\} \leq \frac{f(\boldsymbol{\theta}_1) - f(\boldsymbol{\theta}^*)}{\sum_{t=1}^{T} \eta} + \frac{G^2 L}{2}\frac{\sum_{t=1}^{T} \eta^2}{\sum_{t=1}^{T} \eta}.$$

Since we have implicitly assumed a constant step size, we get the following bound:

$$\min_{t \in \{1,...,T\}} \{\|\nabla f(\boldsymbol{\theta}_t)\|^2\} \leq \frac{f(\boldsymbol{\theta}_1) - f(\boldsymbol{\theta}^*)}{T\eta} + \frac{G^2 L\eta}{2}.$$

If we want the error to be below $\epsilon$, then

$$\frac{f(\boldsymbol{\theta}_1) - f(\boldsymbol{\theta}^*)}{T\eta} + \frac{G^2 L\eta}{2} \leq \epsilon,$$

Therefore error at $T$ is $O(1/T) + O(\eta)$, meaning that we get sub-linear convergence up to some constant that depends on the learning rate. If the learning rate is set to be step dependent, then we can remove the constant and get different learning rates.

## C  ADDITIONAL RESULTS

### C.1  DEFINITION OF FOMOH-BP

FoMoH-BP combines forward-mode and reverse-mode to build an optimization routine that provides the step-size for the ground truth gradient obtained from backpropagation. This additional step sets both tangent vectors in a second-order forward-mode pass to $\nabla f(\boldsymbol{\theta})$. The result is an update step that is equivalent to FoMoH-1D, but with the ground truth gradient:

$$\boldsymbol{\theta}' = \boldsymbol{\theta} - \eta \frac{\nabla f(\boldsymbol{\theta}) \cdot \nabla f(\boldsymbol{\theta})}{\nabla f(\boldsymbol{\theta})^\top \nabla^2 f(\boldsymbol{\theta}) \nabla f(\boldsymbol{\theta})} \nabla f(\boldsymbol{\theta}).$$

### C.2  ROSENBROCK FUNCTION

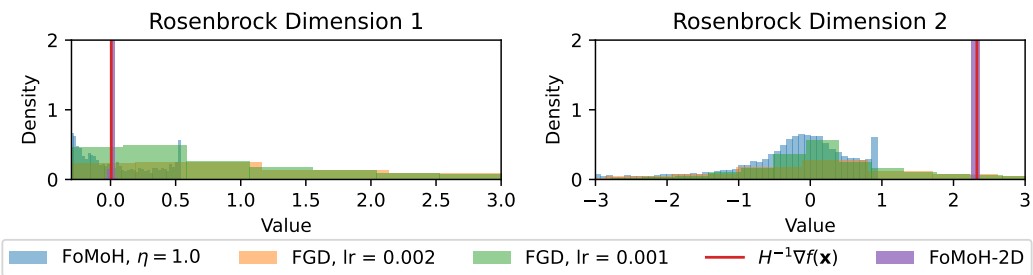

Figure 6: Histogram over expected step taken by the stochastic approaches of FoMoH, FGD, and FoMoH-2D corresponding to Figure 1a. Noteworthy is that the variance of the 2D hyperplane search step is significantly smaller and expectation is close to Newton step.

### C.3  MULTINOMIAL

Logistic Regression

Table 3 includes the final hyperparameter selection for the experimental results in §6.2. We used Biewald (2020) to perform a grid search with 100 iterations, where the batch size choice was between [128, 512, 1024, 2048]. For the learning rate scheduler, we reduced the learning rate at the epoch where the NLL starts to increase. For FoMoH-3D we multiplied the learning rate by $0.8$, whereas for the other approaches we multiplied the learning rate by $0.1$. There is likely room for improvement on the parameters of the learning rate scheduler, but that would only improve the current results.

Figure 7 includes the reverse-mode training and validation curves for Backpropagation and FoMoH-BP in addition to the curves shown in Figure 3.

Table 3: Hyperparameter Optimization for Multinomial Logistic Regression.

| APPROACH | LEARNING RATE | LEARNING RATE BOUNDS | BATCH SIZE |
|---|---|---|---|
| FGD | 0.00006497 | [0.00001, 0.1] | 128 |
| FOMOH-1D | 0.1362 | [0.001, 1.0] | 1024 |
| FOMOH-1D (LR-SCH.) | 0.1362 | [0.001, 1.0] | 1024 |
| FOMOH-2D | 0.04221 | [0.001, 1.0] | 512 |
| FOMOH-2D (LR-SCH.) | 0.04221 | [0.001, 1.0] | 512 |
| FOMOH-3D | 0.1 | [0.001, 1.0] | 512 |
| FOMOH-3D (LR-SCH.) | 0.1 | [0.001, 1.0] | 512 |
| FOMOH-BP | 0.04688 | [0.01, 1.0] | 2048 |
| BACKPROPAGATION | 0.03561 | [0.01, 0.5] | 2048 |

### C.4  CNN

The CNN architecture consists of two convolutional layers, the first with 1 input channel and 20 output channels, the second with 20 input channels and 50 output channels. Both layers have a

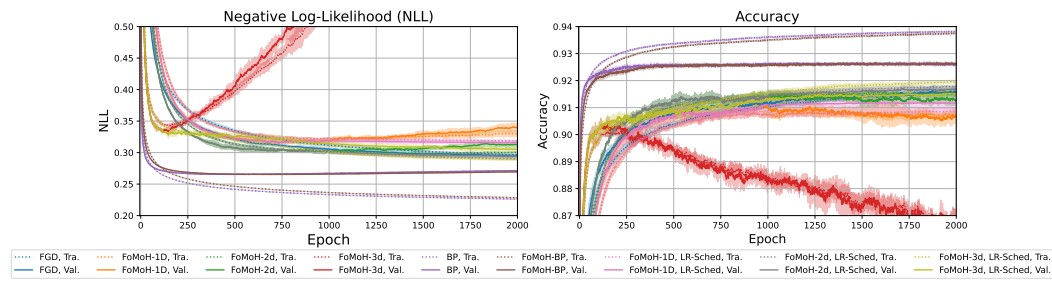

Figure 7: Training and validation curves for multinomial logistic regression model on the MNIST dataset. Average and standard deviation is shown for five random initializations.

kernel size of $5 \times 5$, and each are followed by a ReLU nonlinearity and a $2 \times 2$ 2D max pooling layer. There are then two linear layers. The first has an input of 800 to an output of 500, which is followed by another ReLU, and then a final layer going from 500 to 10.

Table 4 includes the final hyperparameter selection for the experimental results in §6.3. We used Biewald (2020) to perform Bayesian optimization with 100 iterations, where the batch size choice was fixed to 2048. For FoMoH-3D, we used the same hyperparameters as for FoMoH-2D as this gave sufficient performance (and still outperformed the other forward-mode approaches). All learning rate schedulers reduced the learning rate by 10 every 1000 epochs.

Figure 8 includes the reverse-mode training and validation curves for Backpropagation and FoMoH-BP in addition to the curves shown in Figure 4.

Table 4: Hyperparameter Optimization for CNN.

| APPROACH | LEARNING RATE | LEARNING RATE BOUNDS | BATCH SIZE |
|---|---|---|---|
| FGD | 0.0001376 | [0.00001, 0.1] | 2048 |
| FoMoH-1D | 0.542 | [0.001, 1.0] | 2048 |
| FoMoH-1D (LR-SCH.) | 0.542 | [0.001, 1.0] | 2048 |
| FoMoH-2D | 0.3032 | [0.001, 1.0] | 2048 |
| FoMoH-2D (LR-SCH.) | 0.3032 | [0.001, 1.0] | 2048 |
| FoMoH-3D | 0.3032 | [0.001, 1.0] | 512 |
| FoMoH-3D (LR-SCH.) | 0.3032 | [0.001, 1.0] | 2048 |
| FoMoH-BP | 0.04688 | [0.01, 1.0] | 2048 |
| BACKPROPAGATION | 0.03561 | [0.005, 0.2] | 2048 |

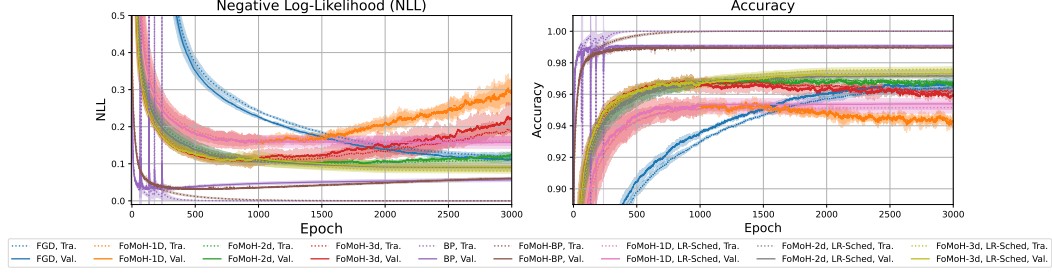

Figure 8: Training and validation curves for CNN on the MNIST dataset. Average and standard deviation is shown for three random initializations.

## C.5 COMPUTATIONAL RESOURCES

The multinomial logistic regression and CNN experiments were run on a NVIDIA RTX 6000 GPU. The main compute cost came from the grid search and the Bayesian optimization for hyperparameter selection.

## D    NESTED FORWARD-MODE AUTOMATIC DIFFERENTIATION

We include a nested forward-mode AD implementation in JAX (Bradbury et al., 2018), since this is also a viable alternative approach to hyper-dual numbers in evaluating the JVP and second-order bilinear form. In the implementation below, hqp_val[0] = $\mathbf{v}_1^\top \nabla^2 f(\boldsymbol{\theta}) \mathbf{v}_2$.

---

**Algorithm 3** Nested forward-mode in JAX (Bradbury et al., 2018)

---

```
import jax
def hqp(fun, w, v1, v2):
    def jvp_v1(w):
        fun_val, jvp_val = jax.jvp(fun, (w,), (v1,))
        return jvp_val, fun_val
    (jvp_val, fun_val), hqp_val = jax.jvp(jvp_v1, (w,), (v2,))
    return fun_val, jvp_val, hqp_val
```

---

## E    WALL-CLOCK TIME

We further include the equivalent plot for Figure 2, where the wall-clock time is now the x-axis. To calculate the wall-clock time, we average over $1,000$ update steps for each $K$ and then multiply the average time per step by each iteration. The result is given by Figure 9. We find that the increased overhead in larger $K$ does not affect the ordering of convergence performance.

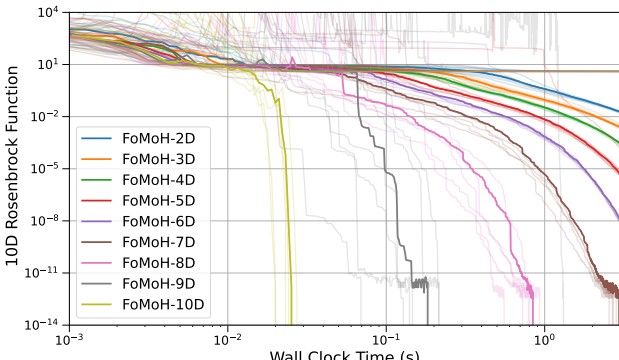

Figure 9: Performance of FoMoH-$K$D for $K = 2 \dots 10$ on the 10D Rosenbrock function against **wall-clock time**. Solid lines represent the median, with transparent lines corresponding to the each of the 10 random seeds. We see that despite the fact that increasing $K$ increases the wall-clock time per step, the trade-off compared to the improved performance is negligible.

