# OpenReview forum: "Second-Order Forward-Mode Automatic Differentiation for Optimization"
_ICLR.cc/2025/Conference — Submitted to ICLR 2025_

### Official Review · Reviewer_qC9Q · 2024-10-29

**Soundness:** 3
**Presentation:** 3
**Contribution:** 2
**Rating:** 3
**Confidence:** 2

**Summary:**

This paper investigates the effectiveness of the second-order differentiation in forward automated differentiation.
This paper and regards the forward propagation with dual number as the objective function applied second-order Taylor-series expansion to.
By using dual numbers, the proposed method estimates the Hessian matrix and applies the approximated Newton's method for the optimization.

**Strengths:**

1. This paper is well-written and easy to follow. Assumptions and mathematical expansions are explained in detail.
1. Experimental results demonstrate the proposed method is more efficient than baseline first-order methods in terms of iterations of parameter updates.
1. Convergence properties are proved in theory, and the proposed method is guaranteed to be consist with Newton's method.

**Weaknesses:**

1. I am not very sure that this paper addresses a new topic in machine learning.
The explanation of the proposed method seems to address general optimization problems but does not seem to focus on the problem in machine learning.
For examples, this paper does not explicitly address the difficulty of accessing the objective function due to large datasets like SGD, and not address the difficulty of non-convexity caused by nonlinear models.
Since I do not have much expertise in optimization theory or operations research, I cannot evaluate the novelty of this paper well in the context of optimization theory.
Even so, I doubt the proposed method is very novel because the used mathematical tools are fundamental and the addressed objective function does not seem very difficult.
Is the research problem specialized for machine learning problems? And, is the paper new even in the context of the optimization problem?


1. While this paper evaluates the convergence of the proposed method in terms of iterations, it does not evaluate the runtime of the proposed method.
The proposed method requires an inverse of Hessian matrices, and I think its computational cost can be high.
Does the overhead of the proposed method not increase the runtime in one iteration? If it does, the proposed method is still faster than baselines in terms of runtime until convergence?
To emphasize the practical usefulness of the proposed method, this paper needs the evaluation of runtime.

1. It is not clear that the proposed method is scalable for recent deep neural network model architectures.
(Fournier et al., 2023) seems to show that the first-order forward gradient method is applicable to ResNet18. Is the proposed method applicable to such modern architectures?

**Questions:**

1. Why is this paper suitable for publication as machine learning research? What is the difficult point of the optimization method in machine learning, and how does the paper address it? If the approximation of a second-order method using dual numbers is new, why has no literature in optimization theory or operations research discussed it?

1. Does not the overhead of the proposed method increase the runtime in one iteration? If it does, the proposed method is still faster than baselines in terms of runtime until convergence?

1. How is the scalability of the proposed method?

---

> ### Author Response · Authors · 2024-11-17
> **Response**
>
> Thanks for your review. We will do our best to respond to your concerns.
>
> > I am not very sure that this paper addresses a new topic in machine learning / Why is this paper suitable for publication as machine learning research?
>
> As we outline in the introduction, using forward-mode automatic differentiation for optimization instead of reverse-mode automatic differentiation is picking up more interest within the machine learning community (See Baydin et al., 2022, Ren et al., 2022; Fournier et al., 2023 etc.). Therefore, we believe this paper adds value to the existing literature by focusing on second-order forward mode AD and how it might be useful for optimization. We believe this work to be the first tackling this problem and therefore think that the empirical results along with the theory are suitable for publication. We also highlight the comments from the other reviewers that believe our work to be a contribution to the community.
>
> > Does the overhead of the proposed method not increase the runtime in one iteration?
>
> Thanks for highlighting this point in regards to runtime. Since we are only inverting Hessians defined on hyperplanes with low dimensions, the increase in runtime does not have a significant effect on performance. To show this, we now include the same plot as in Figure 2 in the appendix, with the x-axis swapped for wall-clock time. Hopefully this addition will help provide an answer to this comment.
>
> > Scalability
>
> Thank you for your comments. This is definitely a direction we would like to explore next.

---

> > ### Comment · Reviewer_qC9Q · 2024-11-18
> > **Thank you for the feedback**
> >
> > Thank you for the feedback. However, my concern about novelty still remains.
> >
> > I am not sure that this paper discusses previous work well. For example, [a] seems to address a similar problem in 2011, and [b] is a lecture note about the second-order automatic differentiation using hyper-dual numbers. [a] and [b] do not seem to be studies in a machine learning community. Thus, there might be significant differences in this paper. However, since this paper does not seem to address the optimization problem specialized for machine learning, I could not understand the novelty of this paper clearly.
> >
> > Do you have convincing reasons why no literature in optimization theory or operations research discussed the problem in this paper? Or, can you provide previous work in broader research areas and discuss the differences?
> >
> > [a] Fike, Jeffrey, et al. "Optimization with gradient and hessian information calculated using hyper-dual numbers." 29th AIAA Applied Aerodynamics Conference. 2011.
> > [b] Fike, Jeffrey A., and Juan J. Alonso. "Automatic differentiation through the use of hyper-dual numbers for second derivatives." Recent Advances in Algorithmic Differentiation. Springer Berlin Heidelberg, 2012.
> >
> > > To show this, we now include the same plot as in Figure 2 in the appendix, with the x-axis swapped for wall-clock time. Hopefully this addition will help provide an answer to this comment.
> >
> > Thank you for the new figure! I understand that FoMoH-KD with high dimensions does not have much larger computational cost than that with low-dimensions. In my understanding, there are no results of FGD in a new figure. Could you compare the proposed method with the baselines?

---

### Official Review · Reviewer_yy3y · 2024-11-05

**Soundness:** 2
**Presentation:** 3
**Contribution:** 3
**Rating:** 6
**Confidence:** 2

**Summary:**

The paper presents a forward-mode-only optimization method that uses second order information on randomly sampled hyperplanes.
The method makes use of (K^2 + K)/2 calls to a "double" forward-mode AD, implemented with hyper-dual numbers, computing along the way the Hessian projected onto the drawn plane. The paper presents theoretical results on convex and quadratic functions that also relates it to the Newton method, alongside some empirical validation on a test function and two learning problems.

**Strengths:**

- The significance of developing a preforming optimization method that does not rely on reverse-mode differentiation is very high, and the proposed method seems to make a concrete step in this direction
- The paper is mostly well written and easy to follow, the notation is clear (although a few passages could be made clearer, see below)
- The work does a good job introducing the concept of dual and hyper-dual numbers, which I expect the community to benefit from
-  The theoretical results clearly show the advantages of the proposed method over FGD, as well as the larger scale experiment.

**Weaknesses:**

- One main promise of forward mode differentiation is to vastly decrease the memory requirement for large models, however the paper does not empirically quantifies the advantage of FoMoH in this regard. It would be nice to include memory footprint comparisons (as well as perhaps a table summarizing the runtime and memory complexity of key algorithms)
- I would have appreciated some larger scale experiment with transformer architectures, e.g. for fine-tuning LLMs, which could be an interesting application of the proposed method
- Some details could be better specified in the main paper:
    -  It is not entirely clear to me by reading the paper if the (hyper-)dual numbers allow for the computation of JVPs and bilinear hessian products by just "tracking the epsilons" while computing the function, or if one has actually to implement the above-mentioned operations (I think it's the first). One "implementation" example would help focus ideas.
    - I'm a bit confused about the origin of Equation (1) right. Do the expressions for the $\kappa$ come from manipulation of the resulting Taylor expansion or is it "set by design" (to mimic Newton updates)? In general, I would have appreciated some more details around lines 227 to 240
   - the learning rate scheduler seems like an important addition for empirical performance, some more details (e.g. which scheduler, how did you choose its hyperparamters) in the main paper would be welcomed.
   - some more discussion on relations between FoMoH-1d, FoMoH-BP and FGD would have been nice. For me it is not immediately clear why FoMoH-1d consistently underperforms FGD on the learning tasks and why FoMoH-BP consistently performs on par with BP.
  - what does BP stand for in the experiments? Is it plain (stochastic?) gradient descent or some other adaptive method?

**Questions:**

1. Is it correct to say that the method performs Newton steps in the sampled subspaces (assuming learning rate being 1)? If not, what's the relationship between the two?
2. the $\kappa_i$'s need not be positive, correct?
3. in the learning tasks, are also examples being smapled (i.e. mini-batch updates?). If so, is the proposed method sensitive to the mini-batch size?
4. Do you think the proposed method could be relevant also for forward-mode gradient-based hyperparameter optimization?

---

> ### Author Response · Authors · 2024-11-17
> **Response**
>
> Thanks very much for your review. We appreciate the highlighted strengths and will do our best to respond to your questions and comments:
>
> > Is it correct to say that the method performs Newton steps in the sampled subspaces?
>
> Correct!
>
> > The $\kappa_i$'s need not be positive, correct?
>
> Correct!
>
> > One implementation example:
>
> Thanks very much for pointing this out. In the code that we will make open-source after review, we have a notebook example where we show this for simple sinusoidal functions. Specifically, we define a hyper-dual number in a notebook and then implement a sine function on the hyper-dual number. We then plot the primal component, as well as the JVP and the Bilinear term, by tracking the epsilons. Hopefully this implementation will help.
>
> > I'm a bit confused about the origin of Equation (1) right.
>
> Thanks for pointing this out. As you noted in a later question, it is set by design to perform Newton updates in the subspace. We added Figure 5 in App. A to try and show this clearer thanks to this comment.
>
> > The learning rate scheduler seems like an important addition for empirical performance … more details in the main paper would be welcomed
>
> Agreed. We have now added this to the main paper, but we still rely on the appendix for the exact parameters.
>
> > What does BP stand for in the experiments? Is it plain (stochastic?) gradient descent or some other adaptive method?
>
> It is plain stochastic gradient descent. We have now explicitly referred to SGD in the paper.
>
> > Do you think the proposed method could be relevant also for forward-mode gradient-based hyperparameter optimization?
>
> Thanks for pointing us to this reference, this could be a really interesting direction to look down. It does look like this work might be relevant to forward-mode gradient-based hyperparameter optimization.
>
> Thanks once again for your detailed review! We have updated the paper accordingly and we hope that our response provided additional clarification.

---

> > ### Comment · Reviewer_yy3y · 2024-12-01
> >
> > Dear authors,
> >
> > thanks for your replies which directly answer my questions. I also have read the other reviews and discussion. While I still find this work very promising, I have decided to lower my score due to the open issues in the proof and the lack of experiments on settings that would truly benefit from the introduced method.

---

### Official Review · Reviewer_VLTw · 2024-11-10

**Soundness:** 2
**Presentation:** 3
**Contribution:** 3
**Rating:** 3
**Confidence:** 5

**Summary:**

The authors present a new optimization method relying on forward mode automatic differentiation (AD). Namely, the authors propose to use second order directional derivatives computed along random directions to precondition stochastic estimates of the gradients obtained by forward mode automatic differentiation along these same random directions. The proposed Forward Mode Second-order Hyperplane Search (FoMoH) interpolates between using an approximate Cauchy stepsize (that is a Cauchy stepsize computed with a quadratic approximation of the objective) along a random direction and an approximate Newton step. The proposed method is shown to outperform a standard forward gradient descent on the Rosenbrock function, a logistic regression problem, and the MNIST image classification task with a CNN.

**Strengths:**

- The idea of exploiting second order directional derivatives is original and could be further explored.
- The proposed method shows clearly superior performance than a simple forward gradient descent.

**Weaknesses:**

Unfortunately the paper does not give justice to the potential of the main idea.
Improvements are necessary and possible:
- The method does not require introducing dual numbers. Computing second order directional derivatives can easily be done with nested forward mode autodiff:
```
import jax

def hqp(fun, w, v1, v2):
  def dir_der_v1(w):
    return jax.jvp(fun, (w,), (v1,))[1]
  return jax.jvp(dir_der_v1, (w,), (v2,))[1]
```
The above may be slightly slower than the implementation with dual numbers (the difference is probably minimal) but it is also much simpler to implement than having to adopt a new kind of automatic differentiation library. If one want the best possible implementation, then Taylor-mode automatic differentiation can also be used but again the benefits are really minor. Algorithm 1 is then unnecessarily complicated when it could be written in just K calls to the above function to define the approximate Hessian. Presenting the algorithm with a simple implementation that any user could recode in at most 50 lines of code in jax or pytorch would greatly improve the potential adoption of the method.
- Unfortunately, the proofs are not rigorous, nor are the claims.
  - Theorem 1 is a corollary of Theorem 2 so no need to present it. Also results like $\lim_{t \rightarrow +\infty} \theta_t = \theta^*$ are meaningless: we don't want to wait the time of the universe to see convergence. Rates like the ones provided in Theorem 2 are relevant.
  - In all claims, detail the setting: what algorithm is used, what is theta_t, what is the expectation taken against etc... You may not do that in the main text by lack of space but at least make sure that the appendix contains a result that details all assumptions.
  - The proof of theorem 2 is unfortunately not well detailed:
    - Please give a detailed proof that $\tilde \theta_{t+1} = \tilde \theta_t + P(\tilde \theta^* - \tilde \theta_t)$. We are dealing with quadratics so the proof should boil down to simple linear algebra. Avoid intuitive arguments, just write the equations one by one showing the results. I personally will refuse this paper to be accepted without detailed proofs.
    - Give a proper reference for the fact that the expectation of a projection matrix defined from Gaussian variables is the scaled identity
    - First line of last set of equations of the proof of theorem 2 should read $\tilde \theta_{t+1} - \tilde \theta^* = (1- K/D)(\tilde \theta_t - \tilde \theta^*)$
    - Please be rigorous when you write expectations. You need to detail each time with respect to which randomness you are taking the expectation. For example at one point you write $\tilde \theta_{t+1} = \mathbb{E}[\tilde \theta_t + P(\tilde \theta^* - \tilde \theta_t)]$ but so then $\tilde \theta_{t+1}$ is not random. Unless you meant $E[\tilde \theta_{t+1}] = $. Such lack of rigor is detrimental to the potential of the idea.
    - Section B.4 contains multiple errors:
      - Reaching a critical point does not imply that you reach a minimum unless you make additional assumptions like convexity.
      - Again be rigorous in the use of expectations, one usually use conditional expectations conditioned on the previous iterate for example.
      - The provided rate is clearly not linear. Consider rereading in details the reference for example.
- Second order methods may generally be very sensitive to the batch-size. It would be great to plot a sensitivity analysis of the method with the batch-size for e.g. a given learning rate.
- Consider another dataset than MNIST. MNIST is well known to be particularly easy and may not reflect potential challenges that the method can have.

**Questions:**

- First it would be great to revise the proofs to make them rigorous.
- Could you add a full mathematical definition of FoMoH-BP?
- What is the logistic regression model? I suppose it is not a regression but a classification problem first? Then if you use 7850 parameters it's probably not a simple linear model but some form of Multi-Layer Perceptron?
- Detail the CNN architecture used in the experiment.
- In the abstract, you mention alternative (orthogonal to be exact) methods for forward gradients. They are never compared in the experiments. It would be great to have them.
- By curiosity how can analog optical systems compute derivatives of intermediate functions to implement forward mode automatic differentiation? The reference provided by the authors does not mention AD, at least from its abstract.
- You mention "linesearch" in line 243 but there is no linesearch at all in the algorithm. What do you mean by linesearch?

---

> ### Author Response · Authors · 2024-11-17
> **Response: Part 1**
>
> Thanks very much for your review. Your comments have been really insightful and have given us a chance to improve components of the paper. In particular the comments on making the proofs more rigorous have directly targeted where we also perceived weaknesses and have significantly helped to strengthen them. I have updated the paper accordingly on OpenReview, but I want to directly highlight the improvements here. We hope this answers your concerns (if any context is missing, in the text below, it should be in the updated paper):
>
> > Please give a detailed proof that $\tilde{\boldsymbol{\theta}}_{t+1}=\tilde{\boldsymbol{\theta}}_t+\mathbf{P}(\tilde{\boldsymbol{\theta}}^* - \tilde{\boldsymbol{\theta}}_t)$:
>
> We start from the FoMoH-$K$D update step (see Eq. 3 in updated paper) applied to the transformed space $\tilde{\boldsymbol{\theta}}$:
> $$\tilde{\boldsymbol{\theta}}_{t+1} =  \tilde{\boldsymbol{\theta}}_t -\mathbf{V}\left(\mathbf{V}^{\top} \nabla^2 f(\tilde{\boldsymbol{\theta}}_t) \mathbf{V}\right)^{-1} \mathbf{V}^{\top} \nabla f(\tilde{\boldsymbol{\theta}}_t)$$
> and then using $\nabla^2 f(\tilde{\boldsymbol{\theta}}_t) = 2\mathbf{I}$, and $\nabla f(\tilde{\boldsymbol{\theta}}_t) = 2\tilde{\boldsymbol{\theta}}_t + \mathbf{b}^{\top}\mathbf{G}\boldsymbol{\Lambda}^{-1/2}$ (these definitions come from the diagonalized quadratic function) gives:
>
> $$\tilde{\boldsymbol{\theta}}_{t+1} =  \tilde{\boldsymbol{\theta}}_t - \frac{1}{2}\mathbf{V}\left(\mathbf{V}^{\top}\mathbf{V}\right)^{-1} \mathbf{V}^{\top} \left( 2\tilde{\boldsymbol{\theta}}_t + \mathbf{b}^{\top}\mathbf{G}\boldsymbol{\Lambda}^{-1/2}\right)$$
>
> Finally, noting that $\tilde{\boldsymbol{\theta}}^* = -\frac{1}{2} \mathbf{b}^{\top}\mathbf{G}\boldsymbol{\Lambda}^{-1/2}$ from setting the gradient to zero, we get
> $$\tilde{\boldsymbol{\theta}}_{t+1}= \tilde{\boldsymbol{\theta}}_t -\frac{1}{2}\mathbf{V}\left(\mathbf{V}^{\top}\mathbf{V}\right)^{-1} \mathbf{V}^{\top} \left( 2\tilde{\boldsymbol{\theta}}_t - 2 \tilde{\boldsymbol{\theta}}^*\right)
>      = \tilde{\boldsymbol{\theta}}_t + \mathbf{P} \left( \tilde{\boldsymbol{\theta}}^* -\tilde{\boldsymbol{\theta}}_t \right)$$
>
> Please see the continuation of this response in the next comment.

---

> > ### Author Response · Authors · 2024-11-17
> > **Response: Part 2**
> >
> > > Give a proper reference for the fact that the expectation of a projection matrix defined from Gaussian variables is the scaled identity:
> >
> > We added appendix B.3 to show this proof which uses the rotational invariance property of the normal distribution and Schur’s Lemma. To highlight the result: Let $\mathbf{R} \in \mathbb{R}^{D\times D}$ be an orthogonal matrix such that $\mathbf{R}\mathbf{R}^{\top} = \mathbf{I}_D$. Then rotating $\mathbf{V}$ gives $\mathbf{V}' = \mathbf{R}\mathbf{V}$. We then use the rotational invariance property of the normal distribution to give $\mathbf{V} {\buildrel d \over =} \mathbf{R}\mathbf{V}$, where $\buildrel d \over =$ denotes equality in distribution. We define a transformed projection matrix:
> > $$
> > \mathbf{P}' = \mathbf{V}'(\mathbf{V}'^{\top}\mathbf{V}')^{-1} \mathbf{V}'^{\top} = \mathbf{R} \mathbf{V}(\mathbf{V}^{\top} \mathbf{R}^{\top}\mathbf{R}\mathbf{V})^{-1} \mathbf{V}^{\top}\mathbf{R}^{\top} = \mathbf{R} \mathbf{P} \mathbf{R}^{\top}.
> > $$
> > Since $\mathbf{V}' {\buildrel d \over =} \mathbf{V}$, then $\mathbf{P} {\buildrel d \over =} \mathbf{P}'$, and $\mathbb{E}[\mathbf{P}'] = \mathbb{E}[\mathbf{R} \mathbf{P} \mathbf{R}^{\top}] = \mathbf{R} \mathbb{E}[\mathbf{P}] \mathbf{R}^{\top} = \mathbb{E}[\mathbf{P}]$. As a result of the last equation, we get the relation $\mathbf{R} \mathbb{E}[\mathbf{P}] = \mathbb{E}[\mathbf{P}]\mathbf{R}$ by post-multiplying by $\mathbf{R}$. Therefore, $\mathbb{E}[\mathbf{P}]$ commutes with all  $\mathbf{R}$ in the orthogonal group. Then using Schur's Lemma $$\mathbb{E}[\mathbf{P}] = c\mathbf{I}_D$$ for some constant $c$.
> > To determine $c$, we take the trace of both sides, and use linearity of expectation to move the trace inside the expectation:
> > $$\mathrm{tr}(\mathbb{E}[\mathbf{P}]) = \mathrm{tr}(c\mathbf{I}_D) $$
> > $$\mathbb{E}[\mathrm{tr}(\mathbf{V}(\mathbf{V}^{\top}\mathbf{V})^{-1} \mathbf{V}^{\top})] =c D$$
> > Then, using the identity $\mathrm{tr}(\mathbf{AB}) = \mathrm{tr}(\mathbf{BA})$ for $\mathbf{A} \in \mathbb{R}^{D\times K}$ and $\mathbf{B} \in \mathbb{R}^{K\times D}$:
> > $$
> >     \mathbb{E}[\mathrm{tr}(\mathbf{V}^{\top}\mathbf{V}(\mathbf{V}^{\top}\mathbf{V})^{-1})] =c D
> > $$
> > $$
> >     \mathbb{E}[\mathrm{tr}(I_K)] = K =c D  \implies c = K/D\notag
> > $$
> > Thus,
> > $$
> > \mathbb{E}[\mathbf{P}] = \frac{K}{D}\mathbf{I}_D
> > $$
> >
> > > Please be rigorous when you write expectations:
> >
> > We have now edited the paper to ensure this. Furthermore, in investigating our proof, we realised that to fully complete it, we needed to transform back from $\tilde{\boldsymbol{\theta}}$ to \boldsymbol{\theta}. Therefore, we included an additional step from Equation 5 in B.2 that bounds the rate of convergence of the expected error according to the condition number of $\mathbf{Q}$. This update should also add more rigour.
> >
> > > Section B.4 contains multiple errors:
> >
> > We have fixed these errors by assuming convexity, by using conditional expectations, and we now refer to the rate as “sub-linear”. Really appreciate you spotting this.
> >
> > We now move to your other comments:
> >
> > >  The method does not require introducing dual numbers…
> >
> > We have now highlighted this in the paper and agree that it is necessary to include to improve the potential adoption of the method. Specifically, we link from the main paper to a new section of the appendix that includes your nested forward-mode implementation in JAX, and cite your review as the reason for its inclusion (see new footnote 3).
> >
> > > Could you add a full mathematical definition of FoMoH-BP
> >
> > We have added a new App. C.1 to include this definition.
> >
> > > What is the logistic regression model?
> >
> > Thanks for highlighting this, we meant to write multinomial logistic regression where the model is given by $\mathrm{sofmax}(\mathbf{WX}+\mathbf{b})$, for $\mathbf{W} \in \mathbb{R}^{10\times784}$, and $\mathbf{b} \in \mathbb{R}^{10}$. We have updated the text.
> >
> > > Detail the CNN architecture used in the experiment
> >
> > We have now added this.
> >
> > > By curiosity how can analog optical systems compute derivatives…
> >
> > Thanks - this is a good question. The referenced paper, Pierangeli et al. 2019, is more to highlight emerging unconventional hardware that might cause us to think about other optimization paradigms. Pierangeli et al. 2019 is a paper on performing computations with light. But another interesting paper that looks to train on unconventional hardware includes Wright et al. 2022 (see https://www.nature.com/articles/s41586-021-04223-6).
> >
> > > You mention "linesearch" in line 243…
> >
> > Thanks for pointing this out. By line search, we mean that we sample a $\mathbf{v}$, which is the descent direction, and then we apply the FoMoH-1D update step to determine how far to move along $\mathbf{v}$.
> >
> > Once again, **we really appreciate the detail in which you went through the paper** and we believe these changes significantly improve the paper.

---

> ### Comment · Reviewer_VLTw · 2024-11-19
> **Refining proofs**
>
> Thanks for the answer. The proof for the expectation of the projection is neat.
> I'm adding some additional comments now so that you can keep on providing details, I may give further answers later.
>
> When I said simple linear algebra I meant that, for  $f(x) = \frac{1}{2} \theta^\top A \theta +  \theta^\top b$, you can simply write the equations as (using that $\theta^* = -A^{-1} b$, $\nabla f(\theta) = A\theta + b$, $\nabla^2 f(\theta) = A$),
> $$
> \theta_{t+1} - \theta^* = (I - V(V^\top A V)^{-1} V^\top) A(\theta_t - \theta^*)
> $$
> Then you can rewrite the above equation as
> $$
> A^{1/2}(\theta_{t+1} - \theta^*) = (I - A^{1/2}V(V^\top A V)^{-1} V^\top A^{1/2}) A^{1/2}(\theta_t - \theta^*)
> $$
> Denoting $\bar \theta_t = A^{1/2}\theta_t$, we get
> $$
> \bar \theta_{t+1} - \bar \theta^* = (I - A^{1/2}V(V^\top A V)^{-1} V^\top A^{1/2}) (\bar \theta_t - \bar \theta^*)
> $$
> Note here the discrepancy between the equation above and the equation you give for $\tilde \theta_t$. In other words on one hand you define $\tilde \theta_t = \bar \theta$ (up to a factor 2), on the other hand you define $\tilde \theta_t$ by the recursion
> $$
> \tilde \theta_{t+1} - \tilde \theta^* = (I - V(V^\top V)^{-1} V) (\tilde \theta_t - \tilde \theta^*).
> $$
> We may have that $E[\tilde\theta_t] = E[\bar \theta_t]$ but it is a bit unclear at first glance. I hope you also see that simple linear algebra makes for simpler arguments than using some "intuitive" change of variables.
>
> Minor details:
> - Remove Theorem 1. It's meaningless and detrimental to the paper right now. Theorem 2 provides the meaningful results.
> - You are considering a strongly convex quadratic, not just a convex quadratic if you want $\mathrm{cond}(Q)$ to be well defined and even the iterations to be well defined (if $Q=0$ the iterations are ill-defined even if it is a convex quadratic).
> - Define a quadratic as $f(\theta) = \frac{1}{2} \theta^\top A \theta +  \theta^\top b$ as above. Choosing $Q$ or $A$ is a question of taste but the $1/2$ factor simplifies a lot the exposition and the reading.

---

> > ### Author Response · Authors · 2024-11-22
> > **Refining Proofs: Response**
> >
> > Thanks for your comments and sorry for the delayed response, I have had to travel this week.
> >
> > > Remove Theorem 1. It's meaningless and detrimental to the paper right now. Theorem 2 provides the meaningful results.
> >
> > We agree and have removed Theorem 1 in the updated paper. Any equations that we referred to in the removed Theorem 1 have now been directly included in the updated proof.
> >
> > > You are considering a strongly convex quadratic
> >
> > Agreed. Thanks for spotting this!
> >
> > > Define a quadratic as $f(\boldsymbol{\theta}) = \frac{1}{2} \boldsymbol{\theta}^{\top}\mathbf{A}\boldsymbol{\theta} + \boldsymbol{\theta}^{\top}\mathbf{b}$...
> >
> > We have now edited the paper to use this formulation to make it read easier. Thanks for this suggestion.
> >
> > We hope that the incorporation of the above suggestions to the proofs have helped further refine the proofs.

---

> ### Comment · Reviewer_VLTw · 2024-11-22
> **Refining proofs: continued**
>
> Thanks, could you please answer my comments about the discrepancy between $\bar \theta_t$ and $\tilde \theta_t$ that I pointed out in the previous comment?
> Said simply, a Newton method is affine invariant, so considering the optimization of $f(A^{-1/2}\theta)$ or $f(\theta)$ amounts to the same iterates up to the change of variable.
> On the other hand, a gradient descent is not affine invariant: optimizing $f(A^{-1/2}\theta)$ or optimizing $f(\theta)$ do not produce the same iterates.
> It is unclear why your proposed method could be affine invariant when it's considering a subspace. Essentially your proof would work if
> $$
> E[A^{1/2} V (V^\top A V)^{-1} V A^{1/2}] = k/d I
> $$
> One can actually run some simulations to test that.
> ```
> import jax
> import jax.numpy as jnp
> import jax.random as jrd
>
> jax.config.update("jax_enable_x64", True)
>
> def test(identity_hessian):
>   n = 10**7
>   d = 4
>   k = 2
>   Vs = jrd.normal(jrd.PRNGKey(0), (n, d, k))
>   if identity_hessian:
>     A = jnp.eye(d)
>     Asqrt = A
>   else:
>     A = jrd.normal(jrd.PRNGKey(1), (d, d))
>     U, _, _ = jnp.linalg.svd(A)
>     S = jnp.diag(jnp.exp(-jnp.arange(0, d, dtype=jnp.float32)))
>     A = U @ S @ U.T
>     Asqrt = U @ jnp.sqrt(S) @ U.T
>
>   def proj(V):
>     return Asqrt @ V @ jnp.linalg.solve(V.T @ A @ V, V.T @ Asqrt)
>
>   projs = jax.vmap(proj)(Vs)
>   got = jnp.mean(projs, axis=0)
>   print(got)
>
> test(identity_hessian=False)
> ```
> You'll see that for `identity_hessian=True` we get $k/d I$ but for `identity_hessian=False` we are far from getting this result. For sure, these are only simulations with $10^7$ samples, but in any case, I would love an actual proof.

---

> > ### Author Response · Authors · 2024-11-22
> > **Refining proofs: Response 2**
> >
> > Thanks, I apologise, I misunderstood the comment. Your observation is completely correct. We also ran similar empirical tests. This is the reason why we now derive bounds on the error that depend on the geometry of the function (i.e. the condition number of $\mathbf{A}$.)
> >
> > To be explicit, we perform the optimization step in the transformed space and then work from the error in transformed space (i.e. starting from $\tilde{\boldsymbol{\theta}}_{t+1} = \tilde{\boldsymbol{\theta}}_t + \mathbf{P} ( \tilde{\boldsymbol{\theta}}^* -\tilde{\boldsymbol{\theta}}_t )$) back to the error in the original space. Therefore starting with the transformed space:
> >
> > $\mathbb{E}[\tilde{\boldsymbol{\theta}}_{t+1} - \tilde{\boldsymbol{\theta}}^*] =  \tilde{\boldsymbol{\theta}}_t +  \frac{K}{D}(\tilde{\boldsymbol{\theta}}^* - \tilde{\boldsymbol{\theta}}_t) - \tilde{\boldsymbol{\theta}}^*$
> >
> > $\|\|\mathbb{E}[\tilde{\boldsymbol{\theta}}_{t+1} - \tilde{\boldsymbol{\theta}}^*]\|\| = \frac{D-K}{D}\|\| \tilde{\boldsymbol{\theta}}_t - \tilde{\boldsymbol{\theta}}^* \|\|$
> >
> > We now transform back to original $\boldsymbol{\theta}$ using $\tilde{\boldsymbol{\theta}} = \boldsymbol{\Lambda}^{1/2}\mathbf{G}^{\top}\boldsymbol{\theta}$:
> > $\|\| \mathbb{E}[\boldsymbol{\Lambda}^{1/2}\mathbf{G}^{\top}(\boldsymbol{\theta}_{t+1} - \boldsymbol{\theta}^*)]\|\| = \frac{D-K}{D}\|\| \boldsymbol{\Lambda}^{1/2}\mathbf{G}^{\top}(\boldsymbol{\theta}_t - \boldsymbol{\theta}^*)\|\|$
> >
> > $\|\|\mathbb{E}[\mathbf{e}_{t+1}]\|\| _{\mathbf{A}} =  \frac{D-K}{D}\|\|\mathbf{e}_t\|\| _{\mathbf{A}}$
> >
> > To get back to the Euclidean norm we use the relation $\sqrt{\lambda_{\text{min}}}\|\|\mathbf{x}\|\| _2\leq\|\|\mathbf{x}\|\| _{\mathbf{A}} \leq \sqrt{\lambda _{\text{max}}}  \|\|\mathbf{x}\|\|,$
> >
> > where $\lambda_{\text{min}}, \lambda_{\text{max}}$ are the maximum and minimum Eigenvalues of $\mathbf{A}$, we have the following bounds on the norm of the errors:
> >
> > For $\mathbf{e}_t$:
> >
> > $\frac{D-K}{D} \sqrt{\lambda _{\text{min}}(\mathbf{A})} \|\|\mathbf{e}_t\|\| _2 \leq \frac{D-K}{D} \|\|\mathbf{e}_t\|\| _{\mathbf{A}} \leq \frac{D-K}{D} \sqrt{\lambda _{\text{max}}(\mathbf{A})}\|\|\mathbf{e}_t\|\| _2$
> >
> > For $\mathbb{E}[\mathbf{e}_{t+1}]$:
> >
> > $\sqrt{\lambda_{\text{min}}(\mathbf{A})} \|\|\mathbb{E}[\mathbf{e}_{t+1}]\|\| _2 \leq \|\| \mathbb{E}[\mathbf{e} _{t+1}] \|\| _{\mathbf{A}} \leq \sqrt{\lambda _{\text{max}} (\mathbf{A})} \|\| \mathbb{E}[\mathbf{e} _{t+1}]\|\| _2$
> >
> > We then use these two inequalities to rearrange to get upper and lower bounds respectively (exact rearrangement is in the paper):
> >
> > $\|\| \mathbb{E}[\mathbf{e} _{t+1}] \|\| _2 \leq \frac{D-K}{D} \frac{\sqrt{\lambda _{\text{max}}(\mathbf{A})}}{\sqrt{\lambda _{\text{min}}(\mathbf{A})}} \|\| \mathbf{e}_t\|\| _2$
> >
> > $ \frac{D-K}{D} \frac{\sqrt{\lambda _{\text{min}}(\mathbf{A})}}{\sqrt{\lambda _{\text{max}}(\mathbf{A})}}\|\|\mathbf{e}_t\|\| _2 \leq \|\|\mathbb{E}[\mathbf{e} _{t+1}]\|\| _2$
> >
> > Leading to the final bounds on the Euclidean norm of the expected error at $t+1$:
> >
> > $\frac{D-K}{D} \left(\sqrt{\mathrm{cond}(\mathbf{A})}\right)^{-1}\|\|\mathbf{e}_t\|\| _2 \leq \|\|\mathbb{E}[\mathbf{e} _{t+1}]\|\| _2 \leq \frac{D-K}{D} \left(\sqrt{\mathrm{cond}(\mathbf{A})}\right)\|\|\mathbf{e}_t\|\| _2$
> >
> > Therefore, to summarise, I was not able to prove the case for the projection matrix when the Hessian is $\mathbf{A}$, and my best solution is to bound the error according to the geometry of $\mathbf{A}$.
> >
> > Once again, appreciate the significant effort that you have put in to improve this paper.

---

> > > ### Comment · Reviewer_VLTw · 2024-11-22
> > > **Refining proofs**
> > >
> > > Please, read carefully my comments. Essentially, you don't have that $e_{t+1} = A^{1/2}\tilde e_{t+1}$, not even in expectation.
> > > Your algorithm is not affine invariant, so running it in one space or the other changes the iterates.
> > >
> > > By the way, you really do not need to introduce the svd of A, using the square root matrix of A is sufficient and clarifies the presentation.

---

> > > > ### Author Response · Authors · 2024-11-26
> > > > **Refining proofs: Response 3**
> > > >
> > > > Thank you. I will incorporate the following into the paper as it shows that going from $\boldsymbol{\theta} \rightarrow \tilde{\boldsymbol{\theta}}$ can be explained better in $\boldsymbol{\kappa}$ and $\tilde{\boldsymbol{\kappa}}$. Essentially, we show there is a linear relationship in expectation between $\boldsymbol{\kappa}$ and $\tilde{\boldsymbol{\kappa}}$, such that a geometric decrease in the error of $\tilde{\boldsymbol{\kappa}}$ means a geometric decrease in the error of the original subspace $\tilde{\boldsymbol{\kappa}}$:
> > > >
> > > > We start from $\tilde{\boldsymbol{\theta}}$ and convert to the corresponding $\tilde{\boldsymbol{\kappa}}$:
> > > >
> > > > $\tilde{\boldsymbol{\theta}} = \mathbf{A}^{-\frac{1}{2}}\boldsymbol{\theta} = \mathbf{A}^{-\frac{1}{2}} \mathbf{V} \boldsymbol{\kappa}$
> > > >
> > > > We note that $\tilde{\boldsymbol{\theta}} = \mathbf{V}\tilde{\boldsymbol{\kappa}}$, therefore:
> > > >
> > > > $\mathbf{V}\tilde{\boldsymbol{\kappa}} = \mathbf{A}^{-\frac{1}{2}} \mathbf{V} \boldsymbol{\kappa}$
> > > >
> > > > $(\mathbf{V}^{\top} \mathbf{V})\tilde{\boldsymbol{\kappa}} = \mathbf{V}^{\top}\mathbf{A}^{-\frac{1}{2}} \mathbf{V} \boldsymbol{\kappa} \implies \tilde{\boldsymbol{\kappa}} = (\mathbf{V}^{\top} \mathbf{V})^{-1} \mathbf{V}^{\top}\mathbf{A}^{-\frac{1}{2}} \mathbf{V} \boldsymbol{\kappa}$
> > > >
> > > > Using $\tilde{\boldsymbol{\kappa}}$, we write the FoMoH-$K$D update step:
> > > >
> > > > $\mathbf{V}\tilde{\boldsymbol{\kappa}}_{t+1} =  \mathbf{V}\tilde{\boldsymbol{\kappa}}_t -\mathbf{V}\left(\mathbf{V}^{\top} \nabla^2 f(\tilde{\boldsymbol{\theta}}_t) \mathbf{V}\right)^{-1} \mathbf{V}^{\top} \nabla f(\tilde{\boldsymbol{\theta}}_t)$
> > > >
> > > > $\mathbf{V} \Delta \tilde{\boldsymbol{\kappa}}= -\mathbf{V}\left(\mathbf{V}^{\top} \nabla^2 f(\tilde{\boldsymbol{\theta}}_t) \mathbf{V}\right)^{-1} \mathbf{V}^{\top} \nabla f(\tilde{\boldsymbol{\theta}}_t)$,
> > > >
> > > > where $\Delta \tilde{\boldsymbol{\kappa}} = \tilde{\boldsymbol{\kappa}}_{t+1} -\tilde{\boldsymbol{\kappa}}_t$, then
> > > >
> > > > $\mathbf{V}^{\top} \mathbf{V} \Delta \tilde{\boldsymbol{\kappa}} = -\mathbf{V}^{\top} \mathbf{V}\left(\mathbf{V}^{\top} \nabla^2 f(\tilde{\boldsymbol{\theta}}_t) \mathbf{V}\right)^{-1} \mathbf{V}^{\top} \nabla f(\tilde{\boldsymbol{\theta}}_t)$
> > > >
> > > > $\Delta \tilde{\boldsymbol{\kappa}} = - \left(\mathbf{V}^{\top} \nabla^2 f(\tilde{\boldsymbol{\theta}}_t) \mathbf{V}\right)^{-1} \mathbf{V}^{\top} \nabla f(\tilde{\boldsymbol{\theta}}_t).$
> > > >
> > > > We use $\nabla f(\tilde{\boldsymbol{\theta}}) = \mathbf{V}\nabla g(\tilde{\boldsymbol{\kappa}})$ and $\nabla^2 f(\tilde{\boldsymbol{\theta}}) = \mathbf{V}\nabla^2 g(\tilde{\boldsymbol{\kappa}})\mathbf{V}^{\top}$
> > > > to rearrange the equation above in terms of $\tilde{\boldsymbol{\kappa}}$:
> > > >
> > > > $\Delta \tilde{\boldsymbol{\kappa}} = - \left(\mathbf{V}^{\top} \mathbf{V}\nabla^2 g(\tilde{\boldsymbol{\kappa}}_t)\mathbf{V}^{\top} \mathbf{V}\right)^{-1} \mathbf{V}^{\top} \mathbf{V}\nabla g(\tilde{\boldsymbol{\kappa}}_t)$
> > > >
> > > > Finally, we derive $\Delta \tilde{\boldsymbol{\kappa}}$ in terms of $\boldsymbol{\kappa}$, using $\nabla g(\tilde{\boldsymbol{\kappa}}) =  (\mathbf{V}^{\top} \mathbf{V})^{-1} \mathbf{V}^{\top} \mathbf{A}^{-\frac{1}{2}} \mathbf{V}\nabla g(\boldsymbol{\kappa})$ and $\nabla^2 g(\tilde{\boldsymbol{\kappa}}) = (\mathbf{V}^{\top} \mathbf{V})^{-1} \mathbf{V}^{\top} \mathbf{A}^{-\frac{1}{2}} \mathbf{V}\nabla^2 g(\boldsymbol{\kappa}) \mathbf{V}^{\top} \mathbf{A}^{-\frac{1}{2}} \mathbf{V} (\mathbf{V}^{\top} \mathbf{V})^{-1}$:
> > > >
> > > > $\Delta \tilde{\boldsymbol{\kappa}} = - \left(\mathbf{V}^{\top} \mathbf{V} (\mathbf{V}^{\top} \mathbf{V})^{-1} \mathbf{V}^{\top} \mathbf{A}^{-\frac{1}{2}} \mathbf{V}\nabla^2 g(\boldsymbol{\kappa}) \mathbf{V}^{\top} \mathbf{A}^{-\frac{1}{2}} \mathbf{V} (\mathbf{V}^{\top} \mathbf{V})^{-1} \mathbf{V}^{\top} \mathbf{V} \right)^{-1} \mathbf{V}^{\top} \mathbf{V} (\mathbf{V}^{\top} \mathbf{V})^{-1} \mathbf{V}^{\top} \mathbf{A}^{-\frac{1}{2}} \mathbf{V}\nabla g(\boldsymbol{\kappa})$
> > > >
> > > > $= - \left( \mathbf{V}^{\top} \mathbf{A}^{-\frac{1}{2}} \mathbf{V}\nabla^2 g(\boldsymbol{\kappa}) \mathbf{V}^{\top} \mathbf{A}^{-\frac{1}{2}} \mathbf{V} \right)^{-1} \mathbf{V}^{\top} \mathbf{A}^{-\frac{1}{2}} \mathbf{V}\nabla g(\boldsymbol{\kappa})$
> > > >
> > > > $ = - (\mathbf{V}^{\top} \mathbf{A}^{-\frac{1}{2}} \mathbf{V})^{-1} \left(\nabla^2 g(\boldsymbol{\kappa})\right)^{-1} \nabla g(\boldsymbol{\kappa})$
> > > >
> > > > $=(\mathbf{V}^{\top} \mathbf{A}^{-\frac{1}{2}} \mathbf{V})^{-1} \Delta \boldsymbol{\kappa} \implies \Delta \boldsymbol{\kappa} = (\mathbf{V}^{\top} \mathbf{A}^{-\frac{1}{2}} \mathbf{V}) \Delta \tilde{\boldsymbol{\kappa}}$
> > > >
> > > > Since $\mathbb{E}[\mathbf{V}^{\top} \mathbf{A}^{-\frac{1}{2}} \mathbf{V}] = \mathrm{tr(\mathbf{A}^{-\frac{1}{2}})}\mathbf{I}_K$, this results in a linear relationship between $\Delta \tilde{\boldsymbol{\kappa}}$ and $\Delta \boldsymbol{\kappa}$. Therefore a geometric decrease in the rate of $\mathbb{E}[\tilde{\boldsymbol{\kappa}} - \tilde{\boldsymbol{\kappa}}^*]$ implies a geometric decrease in the original space $\mathbb{E}[\boldsymbol{\kappa} - \boldsymbol{\kappa}^*]$ due to the proportionality.

---

> > > > > ### Comment · Reviewer_VLTw · 2024-11-26
> > > > > **Refining proofs: continued**
> > > > >
> > > > > 1. Define $\tilde \kappa$. You cannot assert $\tilde \theta = V\tilde \kappa$ as long as you have not defined $\tilde \kappa$. And please, do not use words. Use equations referring to $\theta, A, V$ etc... You also need to introduce subscripts.
> > > > > 2. Consider $A = I$ in your proof. Do you get the same rate as the proof you did before?
> > > > > 3. You are getting a rate in $\tilde \kappa_{t+1} - \tilde \kappa_t$. We need a rate in $\tilde \kappa_t - \tilde \kappa_*$
> > > > > I could continue.
> > > > >
> > > > > Recall from my previous comments that, for $\bar \theta_t = A^{1/2}\theta_t$, we have
> > > > > $$
> > > > > \bar \theta_{t+1} - \bar \theta^* = (I - A^{1/2}V(V^\top A V)^{-1} V^\top A^{1/2}) (\bar \theta_t - \bar \theta^*)
> > > > > $$
> > > > > What you need to show then is that
> > > > > $$
> > > > > \|I - E[A^{1/2}V(V^\top A V)^{-1} V^\top A^{1/2}]\|_2 \leq c < 1
> > > > > $$
> > > > > where $\||M\|_2$ denotes the spectral norm of a matrix $M$.
> > > > >
> > > > > This thread is at its 4th message since pointing out some holes in the proof. I won't continue now. The answers provided by the authors are not rigorous. As long as the proofs are not resolved, I will lower my score. I cannot accept a paper with wrong theoretical statements.
> > > > >
> > > > > Moreover, several of my other concerns have not been answered such as comparing to other methods for forward gradients and give sensibilities to batch-size.

---

### Meta-Review · Area_Chair_KCW5 · 2024-12-22

**Metareview:**

The paper introduces a new second-order forward-mode automatic differentiation method for optimization. It generalizes a second-order line search to a K-dimensional hyperplane search using hyper-dual numbers. The method is evaluated on the Rosenbrock function, logistic regression, and learning a CNN classifier.

According to the reviewers, the paper is well-written and easy to follow. The proposed method is shown to be more efficient than baseline first-order forward-mode methods in terms of iterations.

However, reviewers pointed out the lack of coverage of related works, that the method is not scalable for recent deep neural network model architectures, and that the paper does not evaluate the runtime of the proposed method. It was also questioned whether dual numbers are really necessary to present the proposed method. More importantly, a reviewer pointed out holes in the proof of Theorem 2 that the authors were unable to fix.

As a result, we believe the paper is not ready for publication yet.

**Additional Comments On Reviewer Discussion:**

- One reviewer meticulously examined the proofs and mathematical derivations in the paper, pointing out several areas where they felt the rigor was lacking. This led to multiple rounds of revisions and clarifications from the authors, particularly concerning the convergence properties of the method and the use of expectations.  However, the authors were unable to fix the issues.
- The reviewers suggested including comparisons with other forward gradient methods to provide a more comprehensive evaluation of the proposed method's performance. The authors responded by clarifying the relationship between their method and existing ones but did not include additional experimental comparisons.
- Concerns were raised about the scalability of the method to larger deep learning models and datasets. The reviewers requested experiments on more complex architectures and larger datasets to demonstrate the practicality of the method for modern deep learning applications.
- Some reviewers questioned the novelty of the proposed method, arguing that similar ideas had been explored in the optimization literature.

---

### Decision · Program_Chairs · 2025-01-22

Reject